# DNA-delivered antibody cocktail exhibits improved pharmacokinetics and confers prophylactic protection against SARS-CoV-2

Elizabeth M. Parzych[1], Jianqiu Du[2,9], Ali R. Ali[1,9], Katherine Schultheis[3,9], Drew Frase[1,9], Trevor R. F. Smith [3,9], Jiayan Cui[2], Neethu Chokkalingam[1], Nicholas J. Tursi [1], Viviane M. Andrade[3], Bryce M. Warner[4], Ebony N. Gary [1], Yue Li[1], Jihae Choi[1], Jillian Eisenhauer[1], Igor Maricic[3], Abhijeet Kulkarni[1], Jacqueline D. Chu[1], Gabrielle Villafana[1], Kim Rosenthal[5], Kuishu Ren[5], Joseph R. Francica[5], Sarah K. Wootton [6], Pablo Tebas[7], Darwyn Kobasa[4,8], Kate E. Broderick[3], Jean D. Boyer[3,10], Mark T. Esser [5,10], Jesper Pallesen [2,10], Dan W. Kulp [1,10], Ami Patel [1,10] & David B. Weiner [1,10] ✉

Monoclonal antibody therapy has played an important role against SARS-CoV-2. Strategies to deliver functional, antibody-based therapeutics with improved in vivo durability are needed to supplement current efforts and reach underserved populations. Here, we compare recombinant mAbs COV2-2196 and COV2-2130, which compromise clinical cocktail Tixagevimab/Cilgavimab, with optimized nucleic acid-launched forms. Functional profiling of in vivo-expressed, DNA-encoded monoclonal antibodies (DMAbs) demonstrated similar specificity, broad antiviral potency and equivalent protective efficacy in multiple animal challenge models of SARS-CoV-2 prophylaxis compared to protein delivery. In PK studies, DNA-delivery drove significant serum antibody titers that were better maintained compared to protein administration. Furthermore, cryo-EM studies performed on serum-derived DMAbs provide the first high-resolution visualization of in vivo-launched antibodies, revealing new interactions that may promote cooperative binding to trimeric antigen and broad activity against VoC including Omicron lineages. These data support the further study of DMAb technology in the development and delivery of valuable biologics.

Severe acute respiratory syndrome coronavirus 2 (SARS-CoV-2), the causative agent of coronavirus disease 2019 (COVID-19), has resulted in a global pandemic with >520 million infections and >6.2 million lives claimed to date[1]. Despite the availability of several highly effective vaccines and therapeutics, the continued development of countermeasures is needed to support these effort[2,3]. Monoclonal antibody (mAb) therapy has emerged as a clinically valuable tool for the prevention and/or treatment of infectious diseases, including SARS-CoV-2[4].

Numerous antibody formulations targeting the SARS-CoV-2 spike (S) protein have been developed, several of which received emergency use authorization (EUA). Current options include monotherapy Bebtelovimab (LY-CoV1404; Eli Lilly)[5] and the cocktail Evusheld (formerly AZD7442, composed of Tixagevimab/AZD8895+Cilgavimab/AZD1061; AstraZeneca)[6–8], both of which remain active against the SARS-CoV-2 VoC to date[5,9–20]. While Bebtelovimab is indicated for the treatment of mild to moderate disease, Evusheld is uniquely approved for pre-exposure prophylaxis; indeed, AZD7442 reduced the risk of

symptomatic COVID-19 by 77% among high-risk trial participants (NCT04625725). Despite the potential benefit to public health, supply and logistical challenges limit the widespread administration of prophylactic mAbs, particularly in LMIC. Alternative technologies that allow for the simple, rapid, and durable non-IV delivery of such therapies to supplement existing approaches are of great interest.

The DNA-encoded Monoclonal Antibody (DMAb) platform instructs the in vivo expression of functional antibodies using optimized, synthetic plasmids. Avoiding the complex production and limited stability typically associated with protein or lipid-based formulations, this platform allows the efficient delivery of temperature-stable, purified DNA using clinically-validated electroporation technology (CELLECTRA-EP; Inovio Pharmaceuticals) to facilitate uptake and expression[21]. DMAbs against a number of infectious diseases have been described and are protective in animal models[22–26]. Recent advancements have aimed to improve in vivo DMAb expression levels, potency and variant coverage using focused sequence modification, plasmid engineering and multivalent formulations[27–29]. The ability to strategically combine these with additional approaches to generate DMAb(s) with optimal in vivo kinetics and functionality against SARS-CoV-2 would represent a significant advance.

Here we compare protein versions of AZD7442 mAbs with systematically-developed and extensively-characterized DMAbs based on mAb clones COV2-2196 (2196) and COV2-2130 (2130)[30,31], the predecessors of AZD7442[6,7]. Plasmid engineering strategies enhanced in vivo production, resulting in the persistence of 2130 and 2196 DMAbs for over six months. These demonstrate specific binding to SARS-CoV-2 spike, block ACE-2 engagement and mediate potent viral neutralization against all current viral variants, including Omicron lineages. Plasmid delivery confers striking prophylactic protection in multiple murine and hamster challenge models when administered alone or in combination. In comparative studies, DMAbs possess functionality and efficacy indistinguishable from their recombinant counterparts but with superior in vivo longevity relative to rIgG. Furthermore, structural assessment of nucleic acid-delivered antibodies performed on serum-derived 2196 and 2130 DNA-encoded Fabs (dFabs) using cryo-EM reveals the extensive and diverse interactions of each dFab with the SARS-CoV-2 spike trimer, as well as evidence of stabilizing contacts between bound DMAbs to promote antibody cooperativity and broad strain coverage.

## Results

### Plasmid optimization combined with Fc-engineering induces the in vivo production of functionally potent 2130- and 2196-based DMAbs

Anti-SARS-CoV-2 mAb pair COV2-2196 (class I) and COV2-2130 (class III) are human neutralizing Abs (nAbs) that target non-redundant, complementary epitopes within the receptor binding domain of SARS-CoV-2 spike protein (S-RBD). Both epitopes overlap with the ACE-2 binding site to mediate viral neutralization (Supplementary Fig. 1a)[6,30]. They exhibit high antiviral potency and were effective in multiple preclinical SARS-CoV-2 challenge models[7,13,30]. DMAb constructs were designed using publicly available sequences for the variable heavy ($V_H$) and light ($V_L$) domains of mAbs 2130, 2196 and an additional clone, COV2-2381 (2381)[30]. With properties similar to COV2-2196, mAb COV2-2381 also mediated protection against SARS-CoV-2 in large animal models. These were DNA and RNA optimized to promote in vivo transcript production/processing and inserted, along with a wildtype human IgG1 framework (WT), into a verified custom mammalian expression vector. These were generated as single or dual plasmid designs in which the heavy chains and light chains for each clone were encoded on the same (single) or separate (dual) plasmids (Supplementary Fig. 1b, c).

Studies were conducted to determine the relative expression profiles of single vs dual plasmid constructs following facilitated in vivo delivery to wildtype mice via intramuscular injection and electroporation (CELLECTRA-EP)[32] (Fig. 1a, b). Consistent with in vitro studies (Supplementary Fig. 2), dual-plasmid in vivo-delivery led to a 2–4× increase in peak serum DMAb levels compared to single plasmid constructs. These were maintained for at least 6 months post DMAb administration (Fig. 1b) and displayed potent antiviral activity against SARS-CoV-2 pseudotyped virus (USA-WA1/2020) (Fig. 1c). DMAbs 2196(WT) and 2130(WT) were also detected in the bronchoalveolar lavage (BAL) collected from parallel groups of mice at D14 post-administration, indicating that in vivo-launched DMAbs are present the site of infection prior to challenge (Fig. 1d).

In addition to WT constructs, variants of each DMAb were generated that contain triple-residue modifications ("TM"; L234F/L235E/P331S) in the Fc domain that ablate FcR and C1q binding[33] as found in AZD7442 (Supplementary Fig. 1d). Corresponding Fc variants showed strong and similar expression profiles in vivo (Fig. 1e) and comparable activity against authentic SARS-CoV-2 (USA-WA1/2020) virus (Fig. 1f). The epitope specificity of each DMAb construct was confirmed using modified RBDs containing point mutations K444A and F486A which abrogate binding of mAb clones 2130 and 2196/2381, respectively[6,30] (Fig. 1g). We utilized an established ACE-2 inhibition assay[34] to demonstrate the ability of in vivo-launched DMAbs to efficiently block the binding of spike to hACE-2 (Fig. 1h, i).

Numerous SARS-CoV-2 lineages bearing mutations in the spike protein have emerged, including B.1.1.7/alpha[35], B.1.351/beta[36], P.1/gamma[37], B.1.526/iota[38], B.1.617.2/delta[39] (Supplementary Fig. 3a). Mutations in S-RBD are more likely to interrupt binding by neutralizing mAbs and confer therapeutic resistance[40,41]. We evaluated the relative binding of these DMAbs to mutant RBDs compared to the parental RBD (USA-WA1/2020) (Supplementary Fig. 3b). 2130-based DMAbs retained similar recognition of all mutant RBDs while 2196-derived DMAbs showed modest reduction in binding to the E484K single RBD mutant. Consistent with binding assays (Supplementary Fig. 3b), 2130-based DMAbs were equally potent against these early pseudotyped virus variants (<3-fold reduction in ID50) (Fig. 2a–e). 2196-derived DMAbs demonstrated mild (5-to-8 fold) reduction in activity against B.1.351 while retaining activity against other major variants (Fig. 2a–e). Reproducible serum potency of 1–2 ng ml$^{-1}$ was observed in all individual sera samples, which was unaffected by the Fc framework (WT vs TM) (Fig. 2f, g and Supplementary Fig. 3c).

### DMAb prophylaxis protects mice against SARS-CoV-2 (USA-WA1/2020) lethal challenge

In vivo DMAb efficacy was next evaluated using a validated SARS-CoV-2 lethal challenge model (Fig. 3a)[42,43]. K-18 mice were administered DMAb 2196(TM) or DMAb 2130(TM) and serum expression was measured, reaching high and comparable levels of 30–40 μg/mL at the time of challenge (D15) (Fig. 3b). At Day 4 (D4) post-challenge, viral titers in the nasal turbinates (NT; Fig. 3c) and lungs (Fig. 3d) were reduced in DMAb groups compared to control mice; in the lung, this was a similar reduction of >4–6 logs in both DMAb groups. DMAb animals also had decreased lung pathology relative to controls (Fig. 3e) and were protected from progressive weight loss (Fig. 3f). 100% of treated animals survived while all control animals succumbed to infection (Fig. 3g). Efficacy afforded by the DMAb WT variants (2196(WT) or 2130(WT)) was also confirmed in a non-lethal AAV6.2FF-hACE-2-transduced murine challenge model[44] in which DMAb-expressing animals had similar and significant reductions in lung viral burden (1–2 logs) compared to naïve animals (Supplementary Fig. 4). These data demonstrate the ability of in vivo-launched 2130 and 2196-based DMAb variants to mediate viral reduction, prevent lung inflammation/pathology, and protect animals from severe disease and death when administered as monotherapies.

Previous reports demonstrated the complimentary and synergistic nature of mAbs 2130 and 2196[6,30]. We evaluated protective efficacy

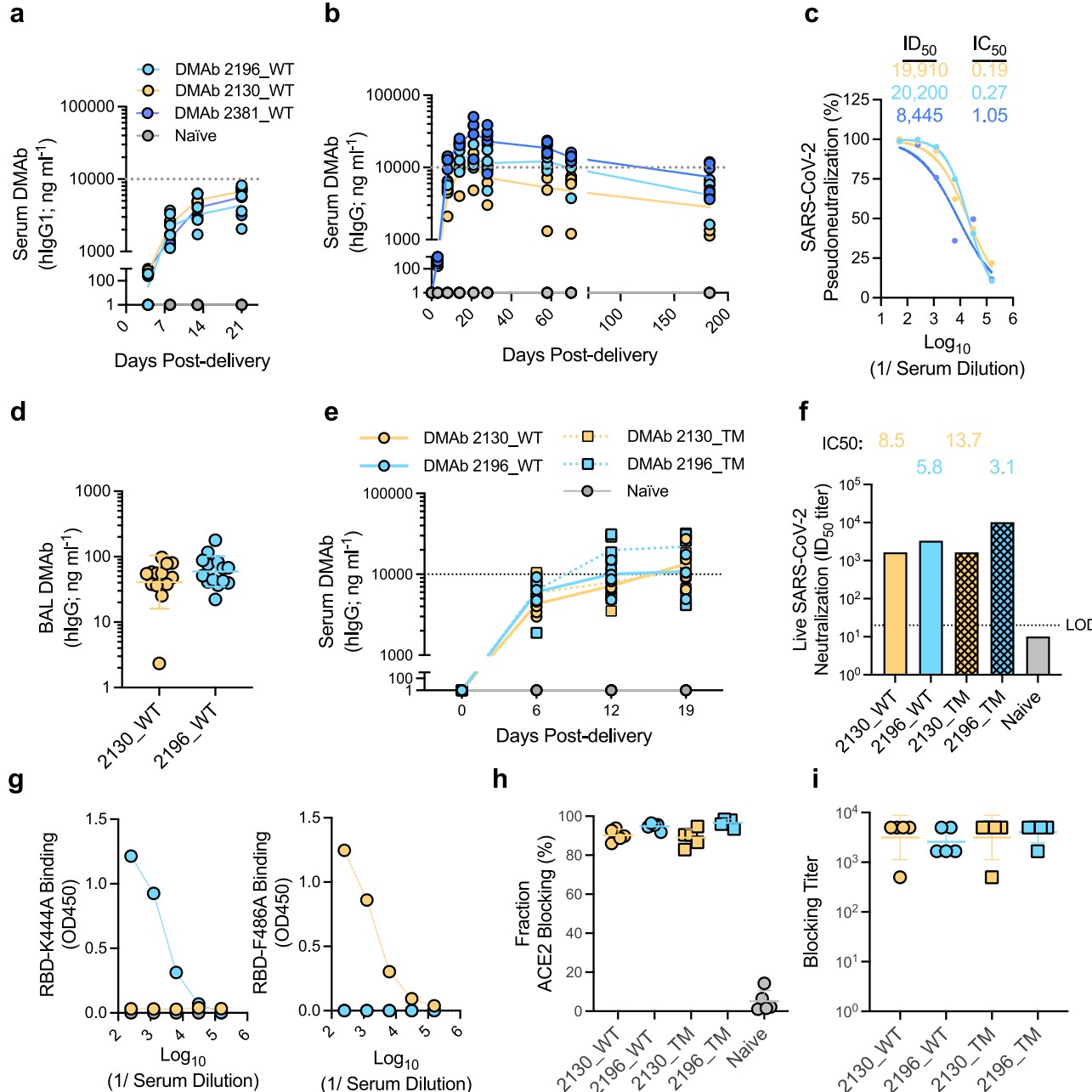

**Fig. 1 | Expression and characterization of in vivo-launched SARS-CoV-2 DMAb constructs.** WT DMAb expression (100 μg dose) following **a** single plasmid ($n = 5$ (2196_WT and 2130_WT) or 4 (2381_WT) independent biological replicates) or **b** dual plasmid constructs ($n = 5$) was measured in the sera of 6–8-week-old female BALB/c mice. Serum DMAb titers for individual mice over time are shown. Lines indicate the group geometric means (GM). **c** Neutralization of pseudotyped SARS-CoV-2 (USA-WA1/2020) by serum pools (from panel **b**). Neutralization curves for each pool are shown (best fit lines and individual data points derived from technical replicates); ID50 and calculated IC50 values are displayed. **d** DMAb levels in the lung bronchoalveolar lavage (BAL) of 6–8-week-old female BALB/c mice at D14 post-plasmid delivery (100 μg dose; $n = 13$ (2130_WT) or 14 (2196_WT) independent biological replicates). Titers for individual mice are shown with group GM (±geometric standard deviation (GSD)) indicated. **e-i** Expression and characterization of WT and TM DMAb constructs in 6–8-week-old female K-18 mice (100 μg dose; $n = 5$). **e** Titers for individual animals (independent biological replicates) over time are shown. Lines indicate the group geometric means (GM)). **f** Neutralizing activity of pooled sera against authentic SARS-CoV-2 virus (USA-WA1/2020). Graph depicts ID50 and calculated IC50 for each pool. LOD = limit of detection. **g** Reactivity of pooled sera against indicated epitope-specific mutant RBDs. Binding curves (OD450) of each pool (average of technical replicates) are shown. **h-i** ACE2 receptor-blocking activity of individual sera samples from **e** ($n = 5$ biological replicates) was determined; **h** proportion (%) of ACE-2 blocking relative to control wells (group mean ± SD) and **i** calculated blocking DMAb titer (GM (±GSD)). Data representative of >2 independent experiments with similar results. Source data are provided as a Source Data file.

following co-delivery of DMAbs 2196(TM) and 2130(TM) (TM DMAb cocktail), as found in AZD7442 (Fig. 3h). Cocktail-treated animals had robust levels (average of 37 μg ml$^{-1}$) of serum DMAbs (Fig. 3i) that recognized *both* epitope-specific RBD mutants (K444A and F486A), indicating concurrent in vivo expression of 2196 and 2130-derived

DMAbs (Fig. 3j). Post-challenge, DMAb-expressing animals had viral titers below the limit of detection in NT (Fig. 3k) and lungs (Fig. 3l), representing a > 5-log reduction in lung viral loads compared to control animals. No SARS-CoV-2-induced lung pathology was observed in DMAb-expressing mice, while most control mice (75%) exhibited

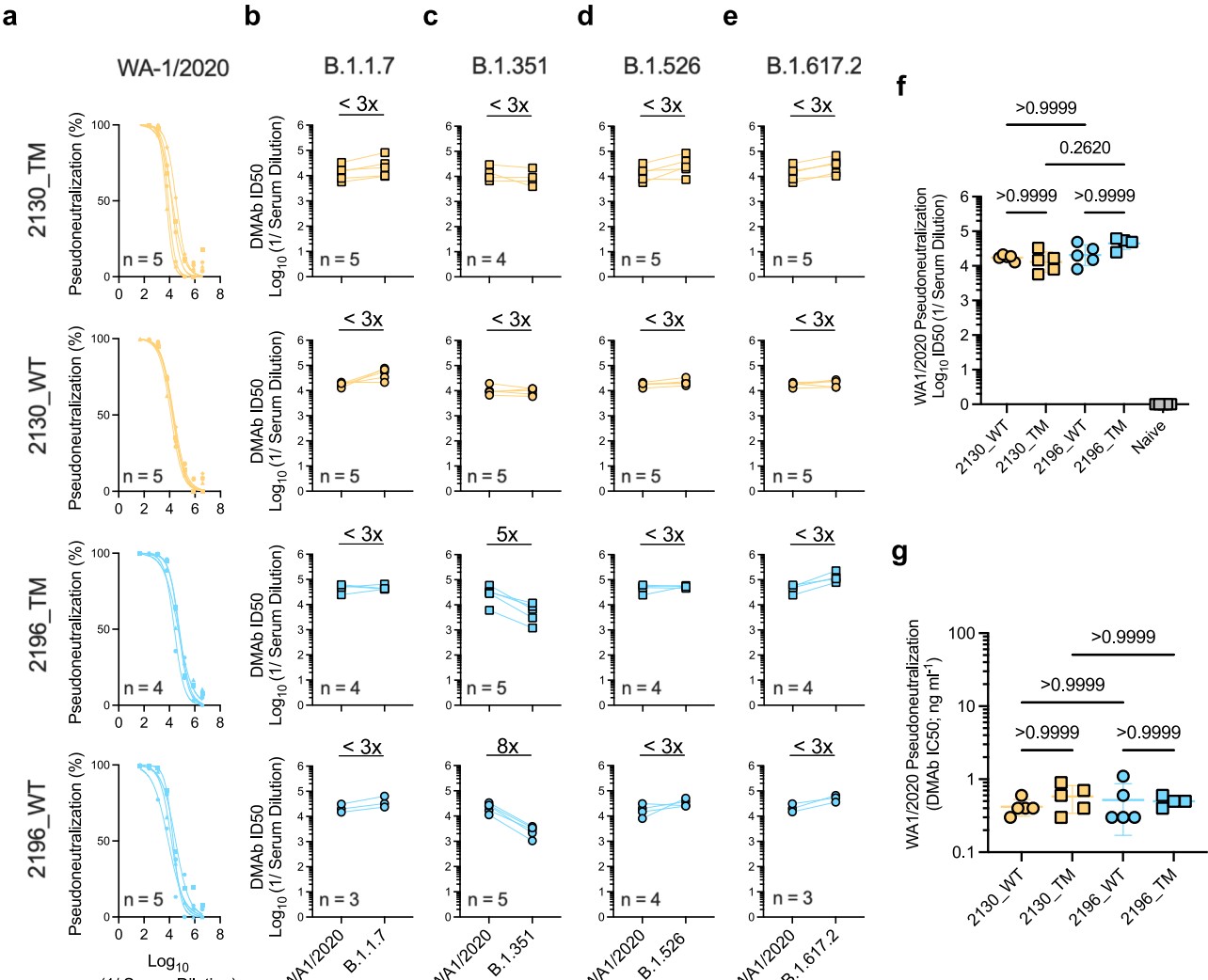

**Fig. 2 | In vivo-launched 2196- and 2130-based DMAbs retain activity against major SARS-CoV-2 viral variants.** Neutralizing activity of sera samples (independent biological replicates) from DMAb-administered mice (n = 3–5/group, as indicated) against **a** USA-WA1/2020 SARS-CoV-2 pseudovirus. Graphs depict neutralization curves for each sera sample (best-fit lines and individual data points derived from technical replicates). **b–e** Corresponding activity of these sera against the indicated variant pseudoviruses. Graphs depict matched ID50s values of each sample against the indicated VoCs compared to USA-WA1/2020. Average fold change (x) in ID50 for each group is indicated for the following SARS-CoV-2 variants: **b** B.1.1.7; **c** B.1.351; **d** B.1.526; **e** B.1.617.2. **f** Comparison of serum ID50 values for individual samples against USA-WA1/2020 (group GM± GSD indicated). Differences between groups were measured using Kruskal–Wallis test followed by Dunn's post hoc analysis. P values indicated. **g** Comparison of calculated IC50 values for individual samples against USA-WA1/2020 (group mean ± SD indicated). Differences between groups were measured using Kruskal–Wallis test followed by Dunn's post hoc analysis. P values indicated. Data reproduced in >2 independent experiments with similar results. Source data are provided as a Source Data file.

inflammation (Fig. 3m). DMAb-treated animals exhibited minimal weight loss (Fig. 3n) and complete survival (Fig. 3o) while progressive weight loss was observed in all control mice leading to significant (88%) death. Pre-challenge sera pools containing the TM DMAb cocktail bound early RBD mutants and maintained neutralizing activity against historical SARS-CoV-2 lineages (Fig. 3p–q). These data verify the in vivo efficacy and functional activity of plasmid-launched DMAbs against multiple SARS-CoV-2 variants following co-delivery.

**Fc-engineered DMAb cocktail(s) exhibit equivalent neutralizing potency and in vivo efficacy relative to protein IgG in murine and hamster challenge models**

Numerous approaches to improve antibody-based therapeutics in patients have been described, including hIgG allotype selection. To facilitate clinical translation, the 2130 and 2196 DMAb plasmids (WT and TM) were modified from the human G1m1 to the G1m3 allotype framework (WT(m3) and TM(m3) constructs) utilized in AZD7442 and

validated in vitro (Supplementary Fig. 5). To directly investigate the potential value of effector engagement in vivo, we conducted an additional efficacy study in K-18 mice comparing TM(m3) or WT(m3) DMAb cocktails (Fig. 4a). As a benchmark standard, an additional group received the WT(m3) rIgG cocktail (purified IgG administered IP). Serum levels of both DMAb cocktails and the rIgG cocktail converged just prior to challenge (Fig. 4b). Both DMAb-administered groups were protected compared to control mice, resulting in >4-log and >2-log reduction in the lung (Fig. 4c) and NT (Fig. 4d), respectively. Viral control between the Fc-modified DMAb groups was indistinguishable from the rIgG cocktail group. Importantly, no immune pathology was detected in the lungs of antibody-treated mice, regardless of variant or mode of delivery (Fig. 4e and Supplementary Fig. 6). Both DMAb cocktails conferred protection against weight loss (Fig. 4f) and mortality (Fig. 4g), compared to naïve control groups (37% survival: 3/8 animals). This protection was as potent as that observed with WT(m3) rIgG cocktail, further validating the in vivo potency and

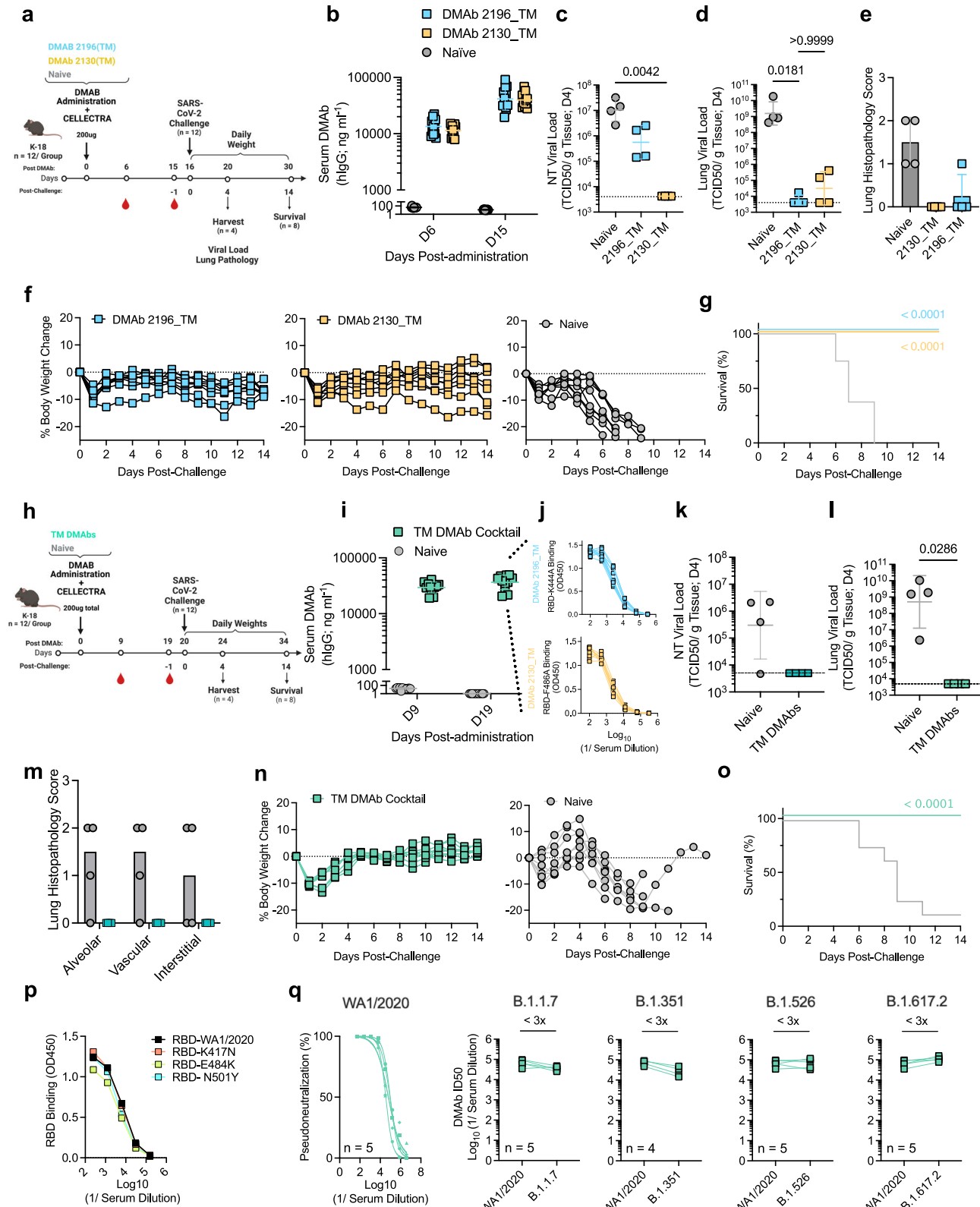

functionality of plasmid-launched antibodies relative to bioprocessed IgG.

The relative in vivo efficacy of effector-engaging and effector-null DMAb cocktails was further validated in a hamster challenge model of SARS-CoV-2 (Fig. 4h). Delivery of DMAb cocktails to Syrian golden hamsters resulted in significant and consistent serum levels (Fig. 4i) of functional antibodies with activity against authentic SARS-CoV-2

(Fig. 4j). Both DMAb cocktails reduced viral loads in the lungs (Fig. 4k) and NT (Fig. 4l), protected against lung pathology (Fig. 4m) and prevented weight loss following challenge (Fig. 4n). Similar to the mouse model (Fig. 4a–g), no significant difference between in vivo expression, in vitro neutralization, body weight loss, pathology or reduction in viral loads were detected between the two DMAb cocktail groups. These data validate the continued efficacy of prophylactic

**Fig. 3 | Prophylactic delivery of 2196_TM and 2130_TM DMAbs protect mice against lethal SARS-CoV-2 challenge. a–g** DMAb prophylaxis against lethal SARS-CoV-2 (USA-WA1/2020) (monotherapy). **a** Schematic of challenge in 6–8-week-old female K-18 mice ($n = 12$ independent biological replicates). **b** Serum DMAb levels in individual animals ($n = 12$) following plasmid delivery (with group GM ± GSD indicated). **c–e** Measurements of viral control (TCID50/g tissue) at D4 ($n = 4$ independent biological replicates). Viral loads (with group GM ± GSD indicated) in the **c**, nasal turbinate (NT) and **d** lung were compared using a Kruskal–Wallis test followed by Dunn's post hoc analysis. $P$ values indicated. Horizontal lines indicate LOD. **e** Histopathology scores for H&E-stained lung sections. Group averages (±SD) and shown. **f**, **g** Challenge outcome in individual animals ($n = 8$ independent biological replicates). **f** Body weight change (%) for each animal and **g** Survival (%) were monitored. Survival curves were compared using a Mantel–Cox Log-rank test. $P$ values indicated. **h–o** DMAb prophylaxis against lethal SARS-CoV-2 (USA-WA1/2020) following co-administration (cocktail). **h** Schematic of lethal challenge in 6–8-week-old male K−18 mice ($n = 12$ independent biological replicates). **i** Serum DMAb levels in individual animals ($n = 12$) are shown. Group GM (±GSD) indicated.

**j** Sera reactivity against epitope-specific mutant RBDs; binding curves (derived from technical replicates) for each animal sample ($n = 8$ biological replicates) are shown. **k−m** Measurements of viral control (TCID50/g tissue) at D4 ($n = 4$ independent biological replicates). Viral load (GM (±GSD) indicated) in the **k** NT and **l** lung were compared using a two-tailed Mann–Whitney $U$ test. $P$ values indicated. Horizontal lines indicate LOD. **m** Histopathology scores for H&E-stained lung sections. Group averages (±SD) are shown. **n**, **o** Challenge outcome in individual animals ($n = 8$ independent biological replicates). **n** Body weight change (%) and **o** Survival (%) were monitored. Survival curves were compared using a Mantel–Cox Log-rank test. $P$ values indicated. **p** Relative binding of pooled sera from cocktail-expressing challenge mice to the indicated mutant RBDs relative to parental D614G RBD; average binding curves for each pool (derived from technical replicates) are shown. **q** Neutralizing activity of individual sera ($n = 5$) against variant pseudo-viruses. Neutralization curves against USA-WA1/2020 (best-fit lines and individual data points derived from technical replicates) are shown. Matched ID50s against the other variants relative to USA-WA1/2020 are shown for each sample. The fold change (x) in ID50 is indicated. Source data are provided as a Source Data file.

DMAb delivery in an additional challenge model of SARS-CoV-2 and suggest that, at these concentrations, additional immune-engagement are not required for protection.

## DNA-delivery further improves the in vivo PK of half-life extending IgG relative to protein administration

To further improve in vivo DMAb half-life, additional variants of WT(m3) and TM(m3) were generated that contain a triple Fc modification (M252Y/S254T/T256E; "YTE") known to promote FcRn-mediated recycling of hIgG into circulation (WT-YTE(m3) and TM-YTE(m3)) (Supplementary Fig. 1e). These constructs were validated in vitro, achieving similar expression levels and antiviral potency as their non-YTE counterparts (Supplementary Fig. 5). The effect of the YTE modification on in vivo DMAb PK was assessed using transgenic mice expressing human FcRn (hFcRn) (Fig. 5). DMAb cocktails containing WT(m3) or WT-YTE(m3) variants were delivered to mice and compared to groups that received corresponding recombinant IgG protein cocktails. Peak levels of recombinant WT mAbs were immediately detected (D1) while groups administered the WT DMAb cocktail had a more gradual accumulation of human IgG in the sera (Fig. 5a, left). Levels in the DMAb and rIgG-treated groups converged between D6-D12. While the amount of rIgG decayed over time, DMAb-treated mice maintained significantly higher levels over the following two months. Parallel groups that received YTE-containing DMAb or protein rIgG cocktails exhibited similar acute kinetics that once again converged by D12 post-administration (Fig. 5a, right) and were maintained at higher levels in the DMAb group over time. YTE-containing constructs display a modestly improvement in titers compared to their non-YTE counterparts, though this does not reach statistical significance. This is the first study to combine in vivo production and YTE function, both of which appear to contribute to sustained DMAb expression to different degrees.

## 2196/2130 cocktails retain recognition and antiviral activity against SARS-CoV-2 Omicron lineages

Sera from hFcRn containing DMAb or rIgG cocktails (±YTE) retained neutralizing activity against USA-WA1/2020 and earlier variants B.1.351 and B.1.617.2 VoC (Fig. 5b). SARS-CoV-2 strain B.1.1.529/BA.1 subsequently emerged with increased transmissibility[45]. Due to its exceptionally high number of spike mutations, BA.1 evades vaccine-induced responses and the majority of clinically-validated mAb therapies[15]. We assessed the reactivity of pooled sera from DMAb-treated hFcRn (Fig. 5c) and BALB/c (Fig. 5d) mice against the BA.1 spike trimer. Specific and similar binding was observed for both DMAb and rIgG groups, indicating that 2196/2130-based cocktails recognize BA.1 (Fig. 5e). This serum also neutralized B.1.1.529/BA.1 pseudotyped virus at ng ml⁻¹ levels, albeit at reduced potency compared to the

USA-WA1/2020 strain (Fig. 5e). Importantly, this activity was largely recovered against the BA.2 variant, comparable to the activity measured against the USA-WA1/2020 strain (Fig. 5f). Consistent with previous studies[10–12,18,20], these data verify the sustained cross-reactivity of 2196/2130 against these dominant and difficult-to-neutralize lineages and support their continued clinical use[46].

## Structural profiling and predictive modeling of in vivo-launched DMAbs

To better visualize the structural profile of in vivo-launched 2196 and 2130-based DMAbs and their interaction with SARS-CoV-2 spike, we performed cryo-EM analysis on serum-derived dFabs. Mice were administered DMAbs 2196 TM(m3) and 2130 TM(m3) in combination or 2196 TM(m3) alone. Total IgG was purified from sera pools, digested and isolated Fabs were complexed with stabilized spike trimer from SARS-CoV-2 (USA-WA1/2020; 6P stabilization) (Supplementary Fig. 7).

Two structures outline the overall interaction of 2196 dFab alone (Fig. 6a–c) or the dFab cocktail of 2196/2130 (Fig. 6d–f) with SARS-CoV-2 spike trimer. A global density map of DMAb 2196/S was generated with a resolution of 3.1 Å. Only one 2196 dFab was present in the 2196/S complex, bound to the single RBD in the "out" position; the other two RBDs were in the "in" position. (Fig. 6a, b). The epitope of 2196 is accessible only in the "out" configuration. Interestingly, the global density map of the 2196/2130/S complex at 3.6 Å revealed the concurrent binding of two 2196 dFabs and three 2130 dFabs per trimer (Fig. 6d–f). Here, two RBDs were in the 'out' position, presenting accessible epitopes to 2196 dFabs while the third maintained the "in" conformation, sterically occluding the binding of 2196. 2130 dFabs bound to all three RBDs regardless of configuration.

As these clones are known to be functionally synergistic, we measured the relative spatial distance between bound dFabs as an indication of their potential to interact with one another. Distance between the center of two 2196 dFabs complexed with spike (Fig. 6d–f; Supplementary Fig. 8a) was 48 Å, allowing both dFabs to recognize spike epitopes simultaneously and facilitating 2196 IgG avidity binding effects. Similarly, 2130 dFabs complexed to RBD 'in' were separated by 50 Å to allow 2130 IgG avidity effects (Supplementary Fig. 8b). The distance between 2130 and 2196 complexed to the same "out" RBD was only 29 Å and the distance between 2130 complexed to 'in' RBD and 2196 complexed to 'out' RBD was 51 Å. Each of these distances allows non-covalent interactions between bound IgGs that contribute to overall cooperative binding to the viral spike trimer.

Subsequent detailed analysis focused on the 2196/2130/spike structure (Fig. 7). Here the paratope/epitope interface of both dFabs with USA-WA1/2020 spike revealed multiple modes of interaction

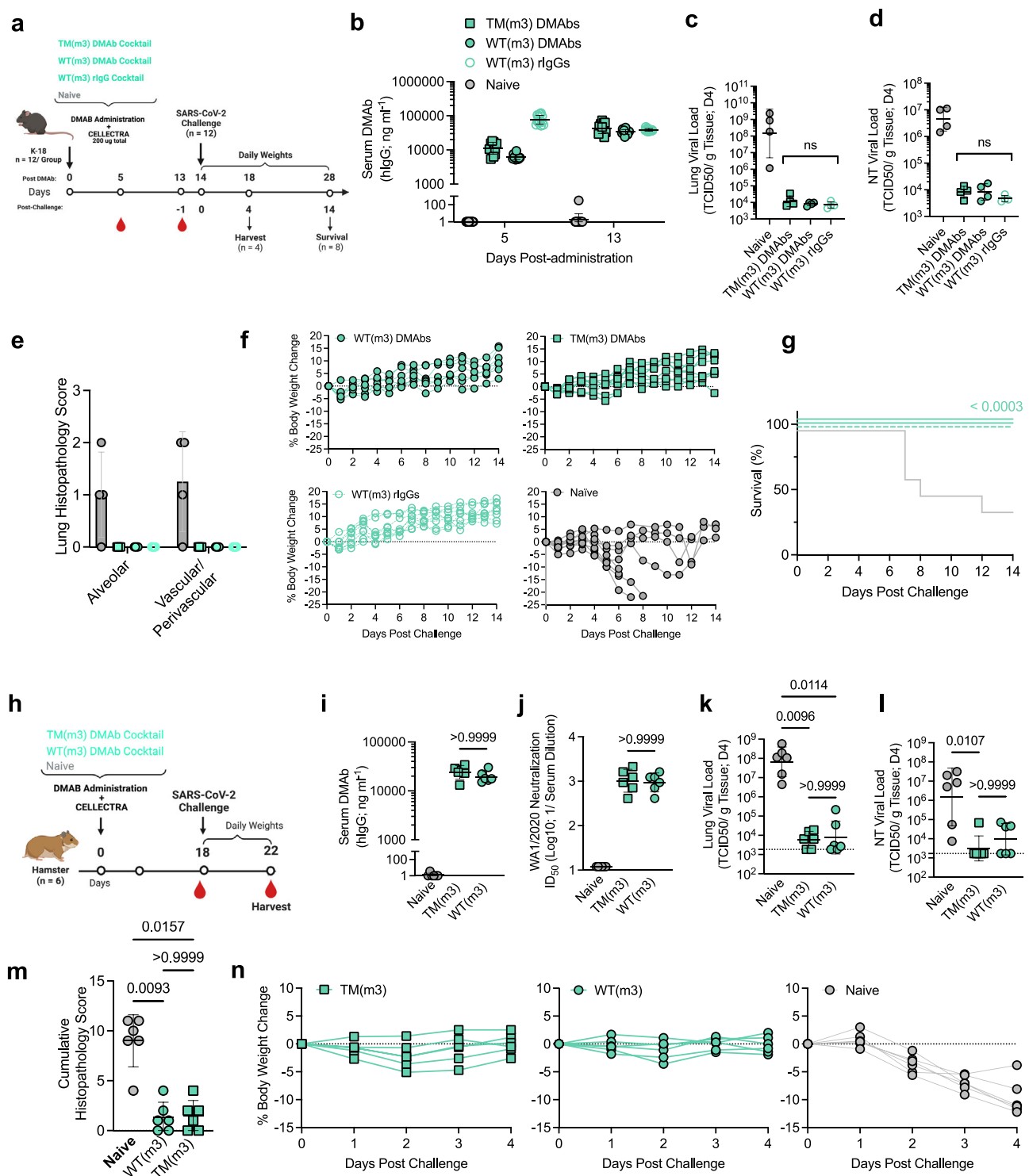

(Fig. 7a) including direct hydrogen bond (h-bond) partners. For dFabs 2130, many of these are relatively resistant to viral mutations since they engage RBD main chain partners, including CDRH3 T102 to RBD R346 peptide bond, RBD 346 to CDRH3 Y100 peptide bond, CDRH3 Y98 top to RBD V445 peptide bond, and RBD N450 to CDRH3 Y100 peptide bond. Other patterns depend on specific side chain interactions, including CHRL1 N30 to RBD S494 and CDRL1 to S30B to RBD 484E (Fig. 7b). H-bonding patterns on the paratope/epitope interface of DMAb 2196 and spike were also extensive, including RBD Q493 with CDRH2 S54, RBD N481 with CDRL1 Y32, RBD N487 with CHRL3 D104 and RBD T478 with CHRL3 D104 (Fig. 7c).

Supporting the notion of dFab cooperative binding effects (Fig. 6d–f), h-bond patterns between the 2196 and 2130 dFabs were observed: 2130 light chain S67 engaged 2196 light chain R95 (Fig. 7b, c), along with additional potential h-bond partners in the vicinity to strengthen this cooperativity. Further support of dFab-to-dFab h-bonding was demonstrated in the density maps. Here we observed increased flexibility (higher than average B factor) of complexes containing the 2196 dFab alone, both in regard to movement of the RBD itself and its interaction with the 2196 dFab (Fig. 6a–c). In contrast, simultaneous binding of both 2196/2130 dFabs results in an ordered and well-defined complex with reduced

**Fig. 4 | Fc-engineered DMAb cocktail(s) confer equivalent protection in both murine and hamster models of SARS-CoV-2 infection, comparable to bioprocessed rIgG. a–g** Efficacy of DMAb cocktails (TM or WT) compared to the rIgG cocktail (WT) benchmark in mice. **a** Schematic of lethal challenge in 6–8-week-old female K-18 mice ($n = 12$ independent biological replicates). **b** Individual serum antibody levels (group GM (±GSD) indicated) following plasmid or rIgG administration ($n = 12$). **c–e** Measurements of viral control in individual animals (TCID50/g tissue) at D4. Viral loads (group GM (±GSD) indicated) in the **c** NT and **d** lung were compared using a Kruskal–Wallis test followed by Dunn's post hoc analysis. $P$ values indicated. Horizontal lines indicate LOD. **e** Histopathology scores for individual H&E-stained lung sections (group averages (±SD) shown). **e, f** Challenge outcome in individual animal ($n = 8$ or 7 (WT(m3) DMAb group) independent biological replicates). **f** Body weight change (%) and **g** Survival (%) were monitored. Survival curves were compared using a Mantel–Cox Log-rank test. $P$ value indicated. **h–n** Efficacy of DMAb cocktails (TM or WT) in a SARS-CoV-2 hamster challenge model. **h** Schematic

of non-lethal challenge conducted in 7–8-week-old female Syrian golden hamsters ($n = 6$ biological replicates). **i** Pre-challenge serum DMAb levels in individual hamsters (group GM (±GSD) shown). **j** Antiviral activity of individual hamster sera (pre-challenge) against live SARS-CoV-2 (USA-WA1/2020). ID50 values (group GM (±GSD)) are displayed. **k–m** Measurements of viral control (TCID50/g tissue) at D4 post-challenge in the **k** lung and **l** NT of individual animals (group GM (±GSD) indicated). Groups were compared using a Kruskal–Wallis test followed by Dunn's post hoc analysis. $P$ values indicated. Horizontal lines indicate LOD. **m** Cumulative lung histopathology score for each animal (group means (±SD) indicated). Lung sections were scored for microscopic indications of edema, hemorrhage, hyperplasia, hypertrophy, metaplasia, mineralization and syncytial cells. Group scores were compared using a Kruskal–Wallis test followed by Dunn's post hoc analysis. $P$ values. **n** Body weight change (%) for each animal following challenge. Source data are provided as a Source Data file.

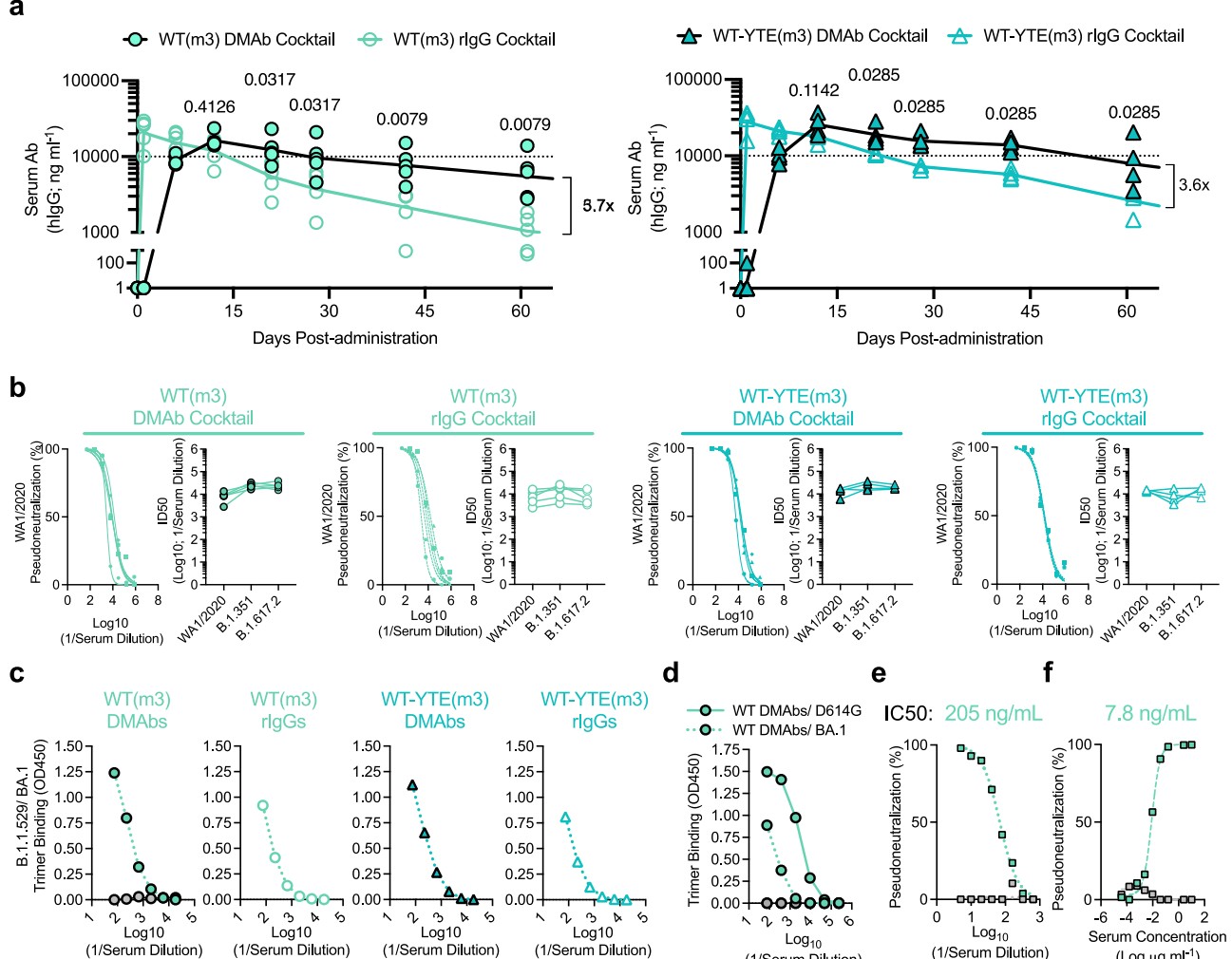

**Fig. 5 | In vivo delivery and half-life engineering (YTE) contribute to improved durability of functional DMAbs compared to bioprocessed rIgG in hFcRn mice.** 6–8-week-old female hFcRn mice ($n = 4$ (WT) or 5 (WT-YTE) independent biological replicates) were administered plasmids encoding the indicated DMAb cocktails (100 μg/animal) or rIgG cocktails (100 μg protein/animal; IP). **a** Individual serum levels (and group GM ± GSD) of the indicated DMAbs or rIgG mAbs are shown ($n = 12$). Group titers at each time point were compared using a two-tailed Mann–Whitney $U$ test. $P$ values indicated. Fold (x) difference in average titers between the indicated groups at D60 post-administration is depicted. **b** Neutralizing activity of all individual hFcRn serum samples ($n = 4–5$/group) against wildtype (USA-WA1/2020) or B.1.351 and B.1.617.2 variant pseudoviruses; neutralization curves against WA1–2020 (best-fit lines and individual data points

derived from technical replicates) are shown, along with matched ID50s against other indicated variants. **c** Relative reactivity of pooled sera from hFcRn mice against recombinant B.1.1.529/BA.1 spike trimer. Average absorbance curves (OD450) displayed (derived from technical replicates). **d–f** Sera harvested from BALB/c mice expressing the DMAb WT(m3) cocktail was pooled for evaluation: **d** Relative reactivity against recombinant spike trimers from B.1.1.529/BA.1 or the parental D614G strains. Average absorbance curves (OD450) of each pool displayed (derived from technical replicates). Naïve serum was used as a control (gray). **e-f** Neutralizing activity of pooled sera (best-fit lines and individual data points derived from technical replicates) against **e** B.1.1.529/BA.1 or **f** B.1.1.529/BA.2 pseudotyped viruses. Calculated IC50 values displayed. Binding and neutralization data has been repeated in >2 independent assays. Source data are provided as a Source Data file.

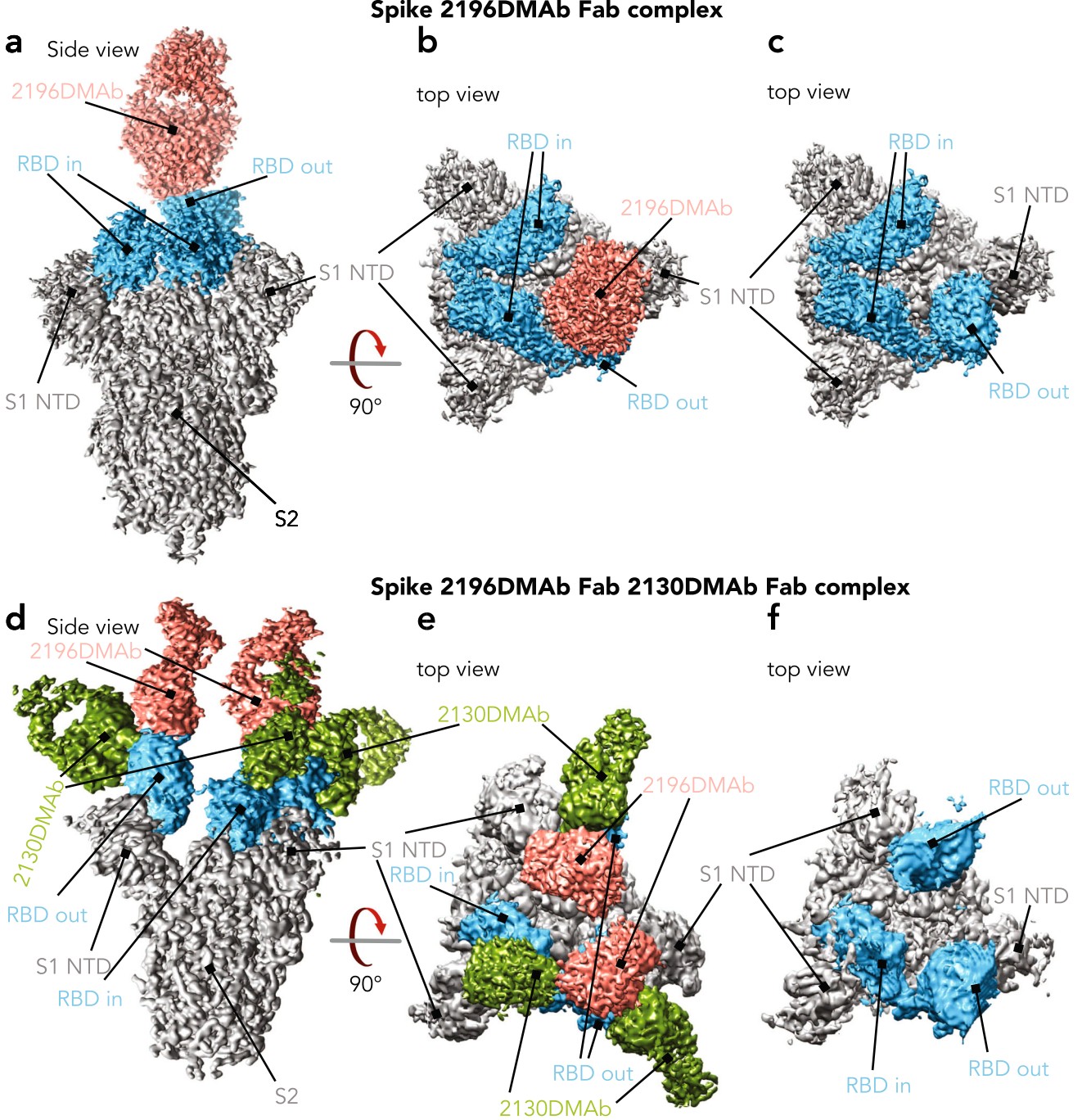

**Fig. 6 | Cryo-EM in vivo-produced dFabs complexed with stabilized SARS-CoV-2 (WA1/2020) spike trimer. a–c** Cryo-EM density map of 2196 dFab (salmon) complexed to SARS-CoV-2 spike trimer (gray) with RBD indicated (blue). **a** Side view. **b** Top view. **c** Top view with dFab density removed. **d–f** Cryo-EM density map of 2196 (salmon) and 2130 (green) dFabs complexed to SARS-CoV-2 spike (gray); spike RBD indicated (blue). **d** Side view. **e** Top view. **f** Top view with dFab densities removed.

motion/flexibility, signifying stabilizing interactions between bound dFabs (Fig. 6d–f).

In addition to h-bonding, both dFabs displayed numerous hydrophobic/van der Waal's interactions with RBD; DMAb 2130 CDRL2 W50 packs against RBD G446, G447 and Y449 (Fig. 7d) and CDRH3 G104-P105 packs against RBD L441 and P499 (Fig. 7e). DMAb 2130 also exhibited multiple cation-pi interactions including CDRL1 30F to RBD Y449; (Fig. 7h) and CDRH3 Y98 to RBD K444 (Fig. 7i). Likewise, dFab 2196 participated in an unusual level of hydrophobic interactions, including the formation of a 5-member hydrophobic cage (composed of CDRL1 Y32, CDRL3 Y91 and W96, CDRH3 P95 and

F106) engaged with RBD F486 (Fig. 7f). Additional hydrophobic interactions of DMAb 2196 include CDRH1 M30, CDRH2 G53 and RBD L455 and L456 (Fig. 7g).

The 2130/2196/S structure was used as a framework to model the impact of recent emerging variants (Supplementary Fig. 9). The B.1.617.2 variant contains a T478K mutation which is relevant to the 2196 epitope (Fig. 7c); however, this did not significantly modulate 2196-mediated neutralization (Fig. 2). Loss of the h-bond between T478 and 2196 CHRH3 D104 (Fig. 7c) could be partly recovered by h-bonding between 2196 CHRH3 D104 and the peptide bond of K478, explaining its maintained activity (Supplementary

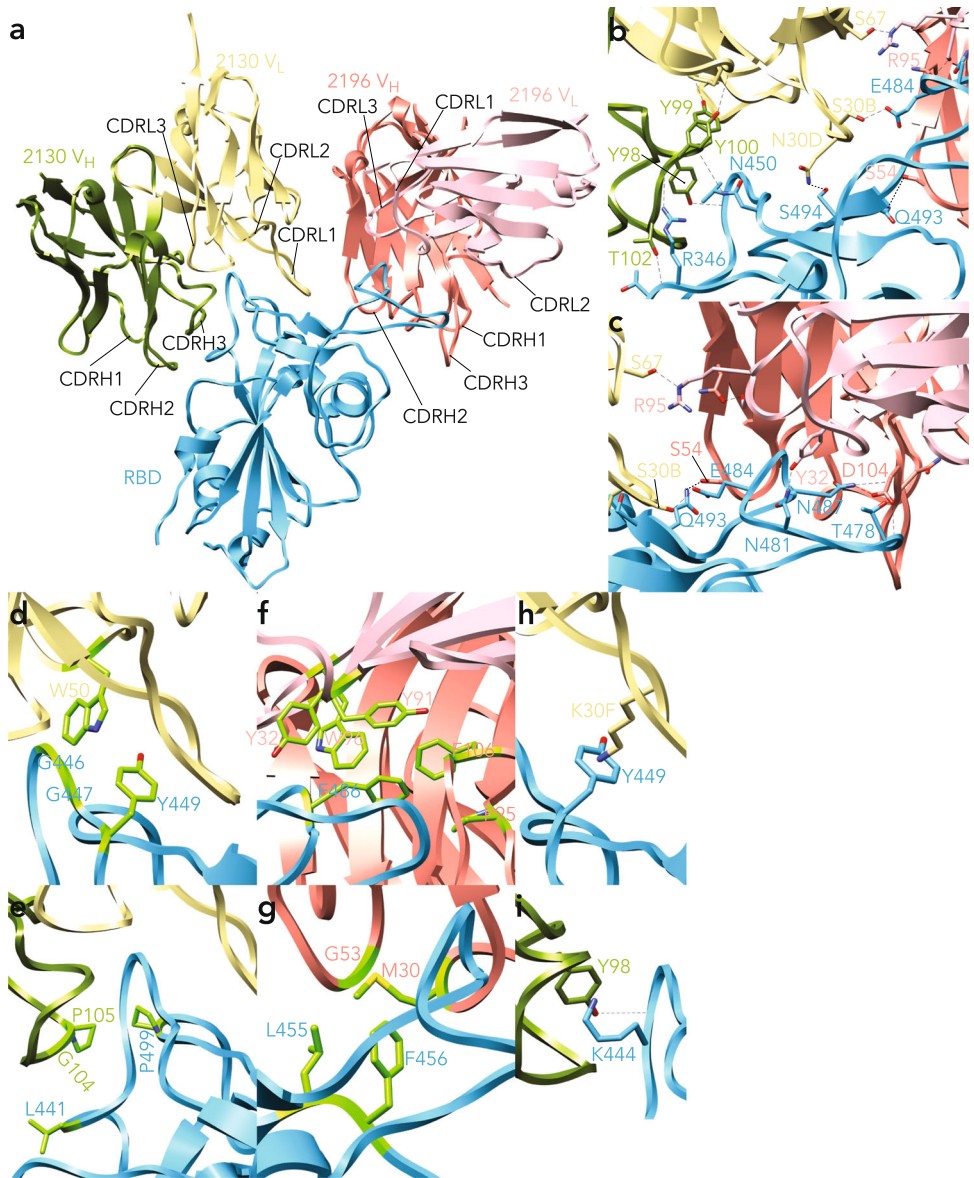

**Fig. 7 | Structural details of the diverse interactions between in vivo-produced dFabs and the SARS-CoV-2 (USA-WA1/2020) spike trimer. a** Structural overview of RBD (blue) in complex with both 2196 ($V_L$ in pink; $V_H$ in salmon) and 2130 dFabs ($V_L$ in gold; $V_H$ in green). **b** 2130 interactions with RBD main chain partners (CDRH3 T102 to RBD R346 peptide bond; RBD 346 to CDRH3 Y100 peptide bond; CDRH3 Y98 top to RBD V445 peptide bond; RBD N450 to CDRH3 Y100 peptide bond) and side chain partners (CHRL1 N30 to RBD S494 and CDRL1 to S30B to RBD 484E). **c** 2196 interactions with RBD (Q493 engages CDRH2 S54, RBD N481 engages CDRL1

Y32, RBD N487 engages CHRL3 D104 and RBD T478 engages CHRL3 D104). **d, e** 2130 interactions with RBD via hydrophobic interactions. **d** CDRL2 W50 packs against RBD G446, G447 and Y449. **e** CDRH3 G104-P105 packs against RBD L441 and P499. **f, g** 2196 interacts with RBD via hydrophobic interactions: **f** hydrophobic cage with RBD F486 formed by CDRL1 Y32, CDRL3 Y91 and W96, CDRH3 P95 and F106. **g** additional hydrophobic contacts include CDRH1 M30, CDRH2 G53 and RBD L455 and L456. **h-i** Cation-pi interactions between 2130 and RBD: **h** CDRL1 30F to RBD Y449. **i** CDRH3 Y98 to RBD K444.

Fig. 9a). The B.1.1.529 variant contains an additional two potentially consequential mutations: Q493R and E484A, both of which engaged in h-bonding in the pre-omicron isolates (Fig. 7c). As the Q493R mutation enables multiple h-bonding partners with the 2196 dFab (Supplementary Fig. 9b; RBD S54 and N56), it is expected to be well-tolerated. Introduction of E484A shortens and changes the chemical properties of the side chain, breaking the hydrogen bond to 2130 CHRL1 S30B (Fig. 7b). However, our binding and neutralization data illustrate that the dFab 2196/2130 cocktail can mitigate this mutation (Fig. 5). These structural studies not only provide the first visualization of in vivo-produced Fabs and their interaction with target antigen, but also support our in vitro and in vivo data indicating that the DMAb 2196/2130 cocktail retains recognition and activity against current SARS-CoV-2 variants.

## Discussion

The COVID-19 pandemic highlighted the value of nucleic acid approaches for the timely development and large-scale deployment of life-saving vaccines. However, the versatility of such platforms extends beyond antigen delivery, potentially allowing the administration of biologically functional therapeutics. Here, we utilized DMAb technology to induce in vivo expression of validated anti-SARS-CoV-2 clones COV2-2196 and COV2-2130 and compared them to the biologic forms[6,30,31]. Pharmacokinetic studies conducted in both BALB/c and K-18 mice demonstrated that optimized expression was achieved with the dual plasmid system, resulting in higher peak DMAb serum titers and long-term expression. In a side-by-side evaluation, DMAbs exhibited prolonged kinetics relative to protein IgG which is a unique advantage of the DNA platform.

We thoroughly characterized their molecular and functional profiles and found DMAbs were comparable to their protein counterparts both in vitro and in vivo[6,30,31]. The potency of in vivo-launched DMAbs against pseudotyped and infectious SARS-CoV-2 (USA-WA1/2020) was high, with $IC_{50}$ values in the low ng ml$^{-1}$ range as previously described[30]. In an AAV6.2FF-hACE-2 model, DMAbs administration reduced lung viral burden by 1–2 logs, a similar degree of control as achieved with a 200 µg dose of the recombinant mAbs in a similar Ad5-hACE2 model[30]. Moreover, prophylactic delivery of DMAbs 2196(TM) and 2196(TM), individually or in combination, conferred complete protection in a lethal mouse model and reduced viral burden in a hamster challenge model by >4 logs. Similar protection from disease and pathology was mediated by both WT(m3) and TM(m3) DMAb cocktails, indicating that Fc-mediated effector mechanisms are not detrimental to their function or safety profiles in these models. Rather, effector functionalities of antibodies may provide potential benefits in immune clearance, particularly at lower levels[47–50]. Importantly, efficacy of the WT(m3) cocktail against lethal challenge was as effective following protein or DNA-delivery, demonstrating equivalency of DMAbs in vivo. The combined strategies of sequence optimization, plasmid engineering and Fc modifications to enhance durability and potency could potentially lower the need for repeated drug delivery.

In addition to kinetic and functional evaluation, we produced the first-ever structures of nucleic acid-delivered, in vivo-produced dFabs. Overall, in vivo analysis supports many of the in vitro structural studies for their protein counterparts. Consistent with initial electron microscopy studies[30], our complexes confirmed that dFab 2130 can recognize its epitope regardless of RBD positioning ('in' or 'out') while dFab 2196 is restricted to the "out" confirmation. Epitope chemistry of each dFab in the 2130/2196 cocktail was characterized, recapitulating essential structural features/interactions previously defined in crystal structures of 2130/2196/RBD complexes[6]. These include the formation of a hydrophobic cage around RBD residue F486 by dFab 2196 involving both heavy and light chain contacts. dFab 2130 demonstrated extensive interactions with key RBD residue K444 with additional interactions noted. We also observed dFab-to-dFab h-bonding between 2196 and 2130 light chains that supported potential interactions previously described[6].

Moreover, we found additional evidence that the two antibodies interact in vivo in a potentially cooperative fashion. Our high-resolution cryo-EM of the full trimeric 2130/2196/S complex revealed the concurrent binding of multiple copies of each dFab, allowing us to visualize and measure the proximity of bound dFabs at nearly full trimer occupancy (5/6 binding sites). Measurements of physical distances support a basis for multiple IgG-to-IgG interactions within the trimer. These data reveal that the cocktail likely benefits from cooperative binding effect via dFab-to-dFab and IgG-to-IgG interactions that could help explain their potency. This, combined with their potential to form compensatory interactions with highly mutated SARS-CoV-2 spike variants such as B.1.617.2 and B.1.1.529/BA.1, could explain their continued activity against SARS-CoV-2 variants. We thus provide a comprehensive understanding of the in vivo-produced 2196 and 2130 dFab cocktail that reveal broader insight into the properties of this valuable clinical mAb pair.

Collectively, this rigorous interrogation of the DMAb approach supports its further study as a prophylactic/immunotherapeutic tool. As a supplement to traditional protein IgG, DNA-delivery could improve the availability of such mAb products by addressing challenges typically associated with the large-scale production, distribution and cold-chain storage of bioprocessed biologics. This could help extend access to underserved populations that may be otherwise restricted due to logistical and/or financial restraints as well as potentially avoid supply chain limitations in the context of future pandemics. Further development and optimization of this technology is likely important.

## Methods

### DNA expression constructs (DMAbs)

The mature variable heavy ($V_H$) and light ($V_L$) domains of the selected mAb clones were optimized at the DNA and RNA levels. Synthetic inserts encoding the heavy chain (HC) and light chain (LC) genes for each clone were designed, containing a leader sequence(s) and the optimized $V_H$ or $V_L$ sequences followed by the corresponding constant domains ($C_H$ and $C_L$, respectively) of wildtype human IgG1 (WT). These were inserted into a modified mammalian expression vector (pVax) under the human cytomegalovirus (hCMV) promoter between an IgG leader sequence and a bovine grown hormone (BGH) polyA signal using single or dual plasmid approaches. In single plasmid constructs (pHC/LC), matching genes were encoded in *cis* and separated by a porcine teschovirus-1 2A peptide/furin cleavage site. For dual plasmid systems, separate light chain plasmids (pLC) and heavy chain plasmids (pHC_WT) were generated for each clone. An additional HC variant, pHC_TM, was generated for selected clones containing a triple mutation (L234F/L235E/P331S) known to nullify effector functions of hIgG1.

### Mammalian cell culture and in vitro transfections

In vitro expression of DNA plasmids was performed in Expi293F™ suspension cells (Thermo Fisher Scientific; A14527). Cells suspension was maintained in Expi293™ Expression Medium (Thermo; A1435101) at 37 °C/8% $CO_2$ conditions and transfected using the Expi293F™ Expression System Kit (Thermo; A14635). All transfection parameters (cell concentrations, culture volumes, DNA dilutions, incubation times, reagent preparations, etc.) were determined according to the manufacturer's guidelines. For in vitro transfection, cells were seeded in 6-well culture plates at $1 \times 10^6$ cells/mL. HC/LC plasmid(s) encoding the indicated DMAbs were diluted in OPTI-MEM media (1 µg ml$^{-1}$; 1:1 ratio) and mixed with EpiFectamine transfection reagent. All constructs were tested in duplicate. DNA:lipid mixtures were incubated for 20 min at room temperature (RT) to allow for complex formation and then added, dropwise, to plated cells. Enhancers were added 18–22 h later, as instructed. Clarified culture supernatants were harvested via centrifugation 4–5 days post-transfection and stored at −20 °C prior to analysis.

### Mice, in vivo DMAb delivery and sample collection

All animal studies were conducted in accordance with federal laws and under protocols approved by the relevant Institutional Animal Care and Use Committee (IACUC). Animal studies were performed in five-to-eight-week-old female BALB/c, male or female K-18 or female hFcRn mice. Transgenic K-18 mice (B6.Cg-Tg(K18-ACE2)2Prlmn/J; 034860; The Jackson Laboratory) express the gene for human angiotensin 1 converting enzyme (hACE2) in the airway epithelia under a human keratin 18 (KRT18) promotor and are susceptible to SARS-CoV-2 infection. hFcRn mice (B6.Cg-Fcgrt$^{tm1Dcr}$Tg(FCGRT)32Dcr/DcrJ; 014565; The Jackson Laboratory) carry a knock-out mutation for mouse Fcgrt and express the gene for human FCGRT under its native hTg32 promotor. This allows a more accurate evaluation of the in vivo kinetics of human IgG. All mice were purchased from certified vendors and housed in The Wistar Institute animal facility. All procedures were performed in accordance with the guidelines from the Wistar Institute Animal Care and Use Committee (IACUC) under approved protocols 201399 or 201464. Animals were housed at an ambient temperature of 20–23 °C and 45–65% relative humidity on a 12 h/12 h light/dark cycle with 15 min transition periods at dusk and dawn. For all DMAb administrations, 50–200 µg of total plasmid DNA was formulated in water supplemented with hyaluronidase (12U/injection; Sigma) and injected into the tibialis anterior(s) and/or quadricep muscle(s). In animals receiving both DMAbs, plasmids for each clone were injected at separate sites. Injections were followed by the delivery of two 0.1 Amp electric constant current square-wave pulses by the CELLECTRA-3P electroporation device (Inovio Pharmaceuticals) to facilitate

plasmid uptake. Recombinant 2196 and 2130 mAbs (100–200 μg per dose) were administered intraperitoneally. To prevent xenogenic responses against human DMAbs, T cell depletion (Anti-CD4⁺/CD8⁺ mAbs, 200 μg per mouse, given intraperitoneally) was performed at the time of plasmid/rIgG injection. For PK studies, sera were periodically collected via submandibular bleed to determine expression levels, durability and functionality. For bronchoalveolar lavage isolation (BAL), animals were euthanized and lungs were flushed with 900 μl of PBS supplemented with 0.05% NaN₃, 0.05% Tween-20, 2% 0.5 M EDTA and protease inhibitor using a 20 G blunt ended needle. BAL fluid was heat-inactivated for 20 min at 56 °C and stored at −20 °C prior to analysis. For efficacy studies, DMAb-treated mice were shipped to collaborators at Public Health Agency of Canada (PHAC) or transferred to BioQual, Inc. for challenge with SARS-CoV-2 (see below). Further experimental details for individual in vivo PK and efficacy studies are indicated in the appropriate Figure(s).

## Western blot
Culture supernatants were probed by Western blot for human IgG expression and presence of the YTE Fc modification. Sample lanes on two identical NuPAGE™ 4–12% Bris-Tris gels (Thermo) were loaded with supernatants containing the indicated DMAbs (200 ng/lane based on ELISA quantification). All samples were reduced with NuPAGE™ Sample Reducing Agent (10X) (Thermo) for 10 min at 70 °C prior to loading. After gel electrophoresis, samples were transferred to PVDF membrane Immobilon -FL (EMD Millipore; IPFL07810) using iBlot™ 2 system (Thermo). Membranes were blocked in OBB (Odyssey® Blocking Buffer; LI-COR) for 1 h and washed with PBS-T (1% Tween-20) and probed with the indicated antibodies. The first gel was probed with mouse anti-beta actin IgG (Sigma; A5316-1000UL; diluted 1:5000 in OBB) as a loading control for 1 h at RT and washed. hIgG DMAbs were visualized using goat anti-hIgG-IRDye-800CW secondary antibody (LI-COR; 926-32232 diluted 1:10,000) in OBB) and bound mouse anti-beta actin was detected with anti-mouse IgG- IRDye-680RD (LI-COR; 926-68070 diluted 1:10,000 in OBB). The second gel was also probed with mouse anti-beta actin IgG as a loading control an anti-YTE IgG monoclonal antibody (AstraZeneca; diluted 1:5000 in OBB) to detect HCs containing the YTE modification for 1 h at RT and washed. YTE-containing HCs and beta actin were visualized with goat anti-mouse IgG- IRDye-RD680, respectively, for 1 h at RT. Finally, membranes were washed three times and scanned using Odyssey® CLx Imager and Image Studio software (LI-COR).

## IgG quantification (anti-human IgG ELISA)
For quantification of DMAb in culture supernatants, NUNC 96-well MaxiSorp plates (Sigma; M9410-ICS) were coated with 5 μg ml⁻¹ goat anti-human IgG-Fc (Bethyl; A80-104A) diluted in 1 × PBS overnight at 4 °C. The following day, plates were washed 4 times with 0.05% PBS-T and were blocked with 5% non-fat dry milk in PBS for 1 h at room temperature (RT). Plates were washed and incubated with duplicate samples, diluted in 1% newborn calf serum (NCS) in 0.2% PBS-T for 1 h at RT. Plates were washed and incubated with 1:10000 HRP-conjugated goat anti-human IgG-Fc (Bethyl; A80-104P) diluted in 1% NCS in 0.2% PBS-T for 1 h at RT. Finally, washed plates were developed with 1-Step™ Ultra TMB-ELISA Substrate Solution (Thermo; 34028) and quenched with 2N H₂SO₄. Plates were read at 450 nm on the BioTek Synergy 2 (Biotek) plate reader. Blank wells were included on each plate and subtracted as background. Purified human IgG (Bethyl; P80-112) was used to create a standard curve for quantification (μg ml⁻¹). Positive control sample was included on each plate and used to standardize values across assays. Data were subsequently exported to Microsoft Excel and analyzed using GraphPad Prism 9. Negative OD values (following background correction) were represented by zero for graphing purposes.

## IgG epitope specificity/variant recognition (antigen-binding ELISAs)
Binding ELISAs were used to confirm the epitope specificities of DMAbs 2130, 2196 and 2381. NUNC 96-well MaxiSorp plates were coated with recombinant RBD proteins (3 μg ml⁻¹ in 1× PBS) containing mutations at residues F444A (RBD-F444A) or F486 (RBD-F486A) (AstraZeneca), which are key residues required for the binding of clones 2130 and 2196/2381, respectively. To evaluate the relative binding of each construct to different VoC, the following coating antigens were used (0.5–1 μg ml⁻¹ in 1× PBS): SARS-CoV-2 Spike RBD-His Recombinant Protein (Sino Biologicals; 40592-V08B), Spike S1(D614G)-His Recombinant Protein (Sino Biologicals; 40591-V08H3), RBD-His K417N Recombinant Protein (Sino Biologicals; 40592-V08H59), RBD-His E484K Recombinant Protein (Sino Biologicals; 40592-V08H84), RBD-His N501Y (Sino Biologicals; 40592-V08H82), Spike S1-K417N/E484K/N501Y/D614G Recombinant Protein (Sino Biologicals; 40591-V08H10), B.1.1.529(BA.1) S1 + S2 Trimer-His Recombinant Protein (Sino Biologicals; 40589-V08H26). ELISA procedure was completed as described above.

## ACE-2 inhibition assays
An established ACE-2 inhibition assay was performed[34]. Briefly, the ability of biotinylated, recombinant ACE2-IgHu to bind plate-bound SARS-CoV-2 RBD protein in the presence of the indicated DMAb(s) was determined. 96-well Flat-Bottom Half-Area plates (Corning) were coated at room temperature for 8 h with 1 μg ml⁻¹ 6x-His tag polyclonal antibody (Thermo; PA1-983B) followed by overnight blocking with blocking buffer containing 5% milk/1× PBS/0.01% Tween-20 at 4 °C. The plates were then incubated with RBD at 1 μg ml⁻¹ at room temperature for 1–2 h. Sera harvested from DMAb-treated mice either were serially diluted 3-fold starting at 1:20 with dilution buffer (5% milk/1× PBS/0.01% Tween-20), added to the plate and incubated at RT for 1–2 h. Human Angiotensin-converting enzyme 2 (ACE2-IgHu) antibody was biotinylated using Novus Biologicals Lightning-Link rapid type A Biotin antibody labeling kit (NovusBio; 370-0010) according to protocol. The biotinylated ACE2-IgHu was added to wells at a constant concentration of 0.5 μg ml⁻¹ diluted with the dilution buffer and incubated at RT for 1 h. The plates were further incubated at room temperature for 1 h with native streptavidin-HRP (Abcam; ab7403) at 1:15,000 dilution followed by addition of TMB substrate (Thermo; 34028), and then quenched with 1 M H₂SO₄. Absorbances at 450 nm and 570 nm were recorded with a BioTek Synergy 2 (Biotek) plate reader. Four washes were performed between every incubation step using PBS and 0.05% Tween-20. The assay was performed in triplicates and the average of the absorbance value was determined. The average absorbance of the lowest dilutions with saturating ACE2 signals was calculated to get a maximum ACE2 binding and no blocking. Each average absorbance value was subtracted from the maximum to get an ACE2 blocking curve. The blocking titer is defined as the reciprocal of the highest dilution where two consecutive dilutions have readings below zero. The maximum area under the curve is determined by calculating the Area Under the Curve (AUC) of full ACE2 binding without the competitor. The AUC of the competitor is then subtracted from the maximum AUC to get the area between the curves (blocking area) and is the measure of ACE2 blocking. The fraction ACE2 blocking is defined as the fraction of the blocking area to the maximum AUC.

## Neutralization assay: pseudotyped virus
For pseudovirus production, HEK293T were maintained in DMEM supplemented with 10% fetal bovine serum (FBS) and 1% penicillin streptomycin (P/S) antibiotic in 37 °C/5% CO₂ conditions. To create SARS-CoV-2 pseudoviruses, Gene jammer (Agilent) was used to transfect cells with 1:1 ratio of pNL4-3.Luc.R-E- plasmid (NIH AIDS reagent) along with various of synthetic plasmids (Genscript) expressing the wildtype spike protein (derived from isolate USA- WA1/2020)

or mutated spikes derived from variants B.1.1.7 (Alpha,) B.1.351 (Beta), P.1 (GammaB.1.617.2 (Delta), B.1.526 (Iota), B.1.1.529/BA.1 (Omicron sublineage BA.1) or B.1.1.529/BA.2 (Omicron sublineage BA.2). Forty-eight hours post-transfection, culture supernatants were collected, enriched with FBS to 12% final volume, and stored at −80 °C. SARS-CoV-2 pseudovirus neutralization assays were established using huCHOAce2 cells (Creative Biolabs; VCeL-Wyb019) plated in a 96-well plate format. Cells were resuspended in D10 media (DMEM supplemented with 10% FBS and 1X Penicillin-Streptomycin), plated (10,000 cells/well) and rested overnight in 37 °C/5% $CO_2$ conditions. The following day, transfection supernatant or sera from DMAb-treated animals were heat-inactivated and serially diluted in duplicate as desired. Supernatant from non-transfected cells or sera from naïve animals served as controls, respectively. Diluted samples were incubated with the indicated SARS-CoV-2 pseudovirus for 90 min at RT and then transferred to rested huCHOAce2 cells. Plates were incubated in 37 °C/5% $CO_2$ conditions for 72 h and then lysed using the britelite plus luminescence reporter gene assay system (Perkin Elmer; 6066769). RLUs were measured using the Biotek Synergy 2 (Biotek) plate reader. Using GraphPad Prism 9, nonlinear regressions were applied to duplicate RLU values for each sample to determine the best fit line. Neutralization titers (ID50) were then calculated, defined as the reciprocal dilution that yielded a 50% reduction in RLU compared to sample control wells; RLUs from cell-only control wells on each plate were subtracted as background prior to analysis. To assess the relative activity against mutant pseudoviruses, the same dilution series was tested in parallel against the indicated variants. The calculated ID50s were used to generate a fold change relative to WA1/2020 ($ID50_{WA1/2020}$ /$ID50_{variant}$). ID50s for each sample were also used along with the corresponding DMAb titer (ng ml$^{-1}$) to calculate inhibitory concentrations (IC50s = DMAb titer/ID50) that reflect the individual molecular potency of a test sample while controlling for expression levels.

## Neutralization assay: authentic SARS-CoV-2 viruses
Live SARS-Related Coronavirus 2, Isolate USA-WA1/2020, was obtained through BEI Resources (NIAID, NIH; NR-52281) and contained within the BSL-3 facility at the Wistar Institute. Vero cells (ATCC; CCL-81) were maintained in DMEM supplemented with 10% fetal bovine serum (FBS). Viral propagation and titration were achieved as previously described. Briefly, the USA-WA1/2020 virus stock was serially diluted in DMEM with 1% FBS and transferred in replicates of 8 to previously seeded Vero cells and incubated for five days under 37 °C/5% $CO_2$ conditions. Individual wells were then scored positive or negative for the presence of cytopathic effect (CPE) by examination under a light microscope. The virus titer (TCID50 ml$^{-1}$) was calculated using the Reed-Munch method and the published Microsoft Excel-based calculator. For neutralization assays, Vero cells were seeded in DMEM with 1% FBS at 20,000 cells/well in 96 well flat bottom plates and incubated overnight. Samples were heat-inactivated at 56 °C for 30 min and then serially diluted in triplicates. These were incubated for 1 h at RT with 300 TCID50 ml$^{-1}$ of virus before the mixture was transferred to previously seeded Vero cells and incubated for 5 days. Neutralizing titers and inhibitory concentration (ID50 and IC50) were determined as described above.

## SARS-CoV-2 challenge: hACE2-AAV model (murine)
Female BALB/C mice ($n = 10$/group) were administered the indicated DMAb plasmid formulations (200 μg/mouse) and shipped to PHAC for evaluation using the previously validated hACE2-AAV model[44]. Mice were anesthetized with isoflurane 14 days post-plasmid delivery and administered $1 \times 10^{11}$ viral copies of AAV6.2FF-hACE2 intranasally (50 μL) to facilitate expression of hACE2 in the lungs of recipient mice. Two weeks later (D21 post-plasmid delivery), mice were given an intranasal challenge with $1 \times 10^5$ TCID$_{50}$ (50 μL) of SARS-CoV-2 virus

(hCoV-19/Canada/ON-VIDO-01/2020; GISAID #EPI_ISL_425177). Controls include a group of non-AAV-transduced animals (insusceptible; negative control) and a group of AAV-transduced/non-DMAb-treated animals (susceptible; positive control). Following challenge, animals were monitored daily for signs of clinical disease and euthanized 4 days post-infection, at which time lung tissue was collected for viral quantification and blood was collected for evaluation of DMAb levels. Levels of viral RNA (copies/g lung tissue) for each animal were determined were determined via qPCR[44].

## SARS-CoV-2 challenge: lethal K-18 model (murine)
Male or female K-18 mice ($n = 12$/group) were treated with the indicated DMAb plasmid formulations (200 μg/mouse) and transferred to BioQual Inc. for evaluation in a lethal SARS-CoV-2 challenge model under IACUC-approved protocol 20–164 (BioQual). Briefly, baseline sera samples and body weights were collected prior to challenge. Mice were anesthetized and intranasally (50 μL) infected with $2.8 \times 10^3$ PFU (SARS-CoV-2/human/USA/WA-CDC-WA1/2020USA_WA1/2020; GenBank Acc: MN985325) (BioQual, Inc.). Animals were monitored daily following challenge for clinical signs of disease (visual scoring, weight-loss, etc.); euthanasia criteria included moribund scoring and/or weight-loss of >20% (vs. pre-challenge starting weight). At D4 post-challenge, a subset of each group ($n = 4$) was sacrificed to assess viral titers in the lungs and nasal turbinates of challenged mice via a validated TCID$_{50}$ assay (BioQual, Inc.). Left lung was collected and placed in 10% neutral buffered formalin for histopathologic analysis. Tissues were processed to hematoxylin and eosin (H&E) stained slides and examined by a board-certified pathologist. Gross and microscopic scoring was conducted, using the following scale that reflects the intensity and pervasiveness of observed histopathological change: Grade 1 (1+): minimal, <10%; Grade 2 (2+): mild, 10–25%; Grade 3 (3+): moderate, 25-75%; Grade 4 (4+): marked, 75–95%; Grade 5 (5+): severe >95%. Additional experimental details for individual K-18 challenges are provided in the appropriate Figure legend(s).

## SARS-CoV-2 challenge: hamster model
7–8-week-old female Syrian golden hamsters were purchased from Envigo (strain HsdHan:AURA). In accordance with institutionally-approved protocol SP2100123 (Inovio), hamsters ($n = 6$/group) were administered the WT(m3) DMAb cocktail (1:1 ratio of DMAb 2130_FcWT(m3) + DMAb 2130_FcWT(m3); 1.6 μg total) or TM(m3) DMAb cocktail (1:1 ratio of DMAb 2130_FcTM(m3) + DMAb 2130_FcTM(m3); 1.6 μg total) intramuscularly followed by CELLECTRA-EP. To prevent xenogenic responses against human DMAbs, T cell depletion (500 μL of 0.7 mg ml$^{-1}$ anti-CD4$^+$/CD8$^+$ mAbs per hamster, given intraperitoneally) was performed 3 days prior to plasmid injection. 18 days post-DMAb delivery, sera were collected and animals were challenged intranasally with SARS-CoV-2 (USA-WA1/2020; GenBank Accession MN985325; 6000 PFU) at BioQual, Inc. under IACUC-approved protocol 20–164 (BioQual). Hamsters were weighed over time and sacrificed 4 days post-challenge (D22) for analysis of viral load in the lung and nasal turbinate tissues via a validated TCID$_{50}$ assay (BioQual, Inc.).

## Cryo-electron microscopy
Serum IgG was recovered from mice that had been administered constructs for in vivo production of either 2196 DMAb or a cocktail of 2130 and 2196 DMAbs. Serum IgG was digested with papain (Sigma; P3125) and Fab was recovered. SARS-CoV-2 6P spike ectodomain peplomers (SARS-CoV-2/human/USA/WA-CDC-WA1/2020USA_WA1/2020; GenBank Acc: MN985325) were expressed in Expi293F™ culture (Thermo Fisher; A14527) and affinity purified via a double strep tag followed by gel filtration using a 10/300 S6I column (Cytiva). dFab and spike peplomer were incubated on ice and complexes purified by S6I gel filtration. Complexes were concentrated in centrifugal filters

(Amicon) and vitrified on 1.2/1.3 gold cryo-electron microscopy grids (Protochips) by use of a Vitrobot Mark IV (Thermo). EFTEM data was collected using a Titan Krios G4 instrument (Thermo) equipped with a Bioquantum K3 detector (Gatan) in electron counting mode. A subset of TEM data was collected on a Talos Arctica equipped with a Falcon 3 detector (Thermo) (Supplementary Table 1). Data collection was automated by use of EPU in AFIS mode (Thermo Fisher Scientific). Dose fractionated movies were recorded of each of the two samples; 7952 movies of Spike/2196 and 9893 movies of Spike/2196/2130. The former data was recorded at nominal magnification of 81,000× (Krios/K3) or 150,000× (Arctica/F3); the latter data at either 81,000× or 64,000× (super resolution). The Spike/2196 complex data treatment was performed in Relion (version 3.1.2)[51]. Movie frame alignment and weighted integration (Relion) were followed by CTF estimation (ctffind4 version 4.1.14)[52]. LoG picking was followed by 2D image classification. Suitable 2D classes were identified and underlying molecular projection images were selected for further processing. A low-pass filtered in-house unliganded SARS-CoV-2 density map was used for initial Euler angle assignment and 3D refinement was conducted. CTF refinement, beam tilt correction and Bayesian polishing were performed resulting in a global density map with a resolution of 3.1 Å (FSC 0.143 criterion). The global density map has two RBDs in the 'in' position and one in the 'out' position; only the 'out' position is occupied by one copy of 2196 dFab. Density subtraction and local refinement focusing on the 'out' RBD/2196DMAb Fab was performed[53,54] resulting in a local density map with a resolution of 4.0 Å (FSC 0.143 criterion). The Spike/2130DMAb_Fab/2196DMAb_Fab data was processed in similar fashion using cryosparc (version 3.3.1)[55]. The global density map has all three RBDs in the 'out' position and each RBD is bound by one copy of 2130 dFab and one copy of 2196 dFab. Resolution of this global density map is 3.6 Å and resolution of the local density map is 4.2 Å (FSC 0.143 criterion). Model building was initiated using the SARS-CoV-2 Spike RBD from PDB 7E23. Models of 2130 dFab and 2196 dFab were obtained using Rosetta's antibody application (version 2021.16.61629)[56] (Chotia numbering); models were adjusted and CDR loops rebuilt manually in Coot (version 0.9.8.3)[57] guided by the local density maps. Atomic models were refined with Rosetta FastRelax (version 2021.16.61629) and geometry evaluated with MolProbity (version 1.19.2) online analysis[58], fit-to-map evaluated with command line implementation of EMRinger (version 1.19.2)[59] and glycan geometry evaluated with Privateer (version MKIV)[60]. Hydrogen bonding patterns were identified using hbonds in UCSF Chimera (version 1.15)[61].

## Statistical analysis

Statistical analyses were conducted using GraphPad Prism 9 software. Nonparametric tests were performed due to small group sizes. Two-tailed Mann–Whitney $U$ tests were used when comparing means of two groups and Kruskal–Wallis nonparametric rank-sum test followed by Dunn's post hoc analysis were conducted to compare three or more groups. Survival curves were analyzed using Mantel–Cox Log-rank test. In all cases, $P$ values < 0.05 were considered significant.

## Reporting summary

Further information on research design is available in the Nature Research Reporting Summary linked to this article.

## Data availability

Publicly available sequences encoding the variable domains of mAb clones COV2-2196 (PDB: 7L7D), COV2-2130 (PDB: 7L7E) and COV2-2381 (PMID 32651581) were used to generate DMAb constructs. The sequence of SARS-CoV-2 Spike protein from isolate USA/WA-2020 (GenBank Accession MN985325) was used to produce the stabilized trimer for cryo-EM studies. Model building was initiated using Spike RBD from PDB 7E23. Refined structures for S/2196 dFab and S/2196

dFab/2130 dFab complexes generated in this study have been deposited in the PDB under accession numbers 8D8R and 8D8Q, respectively. Structures were also submitted to EMDB under codes EMD-27255 and EMD-27254, respectively. Source data are provided with this paper.

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

## Acknowledgements

The authors would like to thank The Wistar Institute Core facilities and The Wistar Vaccine and Immunotherapy Trainee Program (N.J.T. and J.E.). We thank the EM center at IU Bloomington as well as the PI4L consortium and the Purdue cryo-EM facility for assistance in data collection. We thank Jingchuan Sun of PI4L/Purdue for assistance in data

collection and data treatment. This work was funded by the Defense Advanced Research Projects Agency and the Joint Program Executive Office for Chemical, Biological, Radiological and Nuclear Defense (JPEO-CBRN) (Award HR0011-21-9-0001 to D.B.W.). This work was approved for public release, distribution unlimited. The views, opinions and/or findings expressed are those of the author and should not be interpreted as representing the official views or policies of the Department of Defense or the U.S. Government. Additional funding included T32-AI-055400 (to E.M.P.), Award T32 CA09171 (to E.N.G.), the WW Smith Charitable Trust (to D.B.W.), The Jill and Mark Fishman Foundation (to D.B.W.) and the Public Health Agency of Canada (to D.K.). Funding sources were not involved in the design of this study, collection and analyses of data, decision to submit or preparation of the manuscript. Visual schematics were created with BioRender.com.

## Author contributions

D.B.W., A.P. and D.W.K. conceived project and secured funding. E.M.P., A.P., D.B.W., T.R.F.S. D.W.K., J.P. and M.T.E. designed and supervised studies. E.M.P., K.S., A.R.A., D.F., N.C., N.T., V.M.A., B.M.W., E.N.G., Y.L., J.C., J.E., A.K., J.D.C., G.V. generated reagents and performed laboratory experiments. J.P., J.D. and J.C performed structural studies/analysis. K.S., T.R.F.S., I.M., K.Rosenthal., K.Ren., J.R.F., S.K.W., P.T., D.K., K.E.B., J.D.B., M.T.E., provided valuable materials and programmatic support/guidance. E.M.P. prepared the original manuscript. All authors edited, reviewed and approved the manuscript.

## Competing interests

K.S., T.R.F.S., V.M.A., I.M., K.E.B. and J.D.B. are employees of Inovio Pharmaceuticals and as such receives salary and benefits, including ownership of stock and stock options, from the company. J.F, K.R, K.R and M.T.E are employees of and hold or may hold stock in AstraZeneca. S.K.W. is an inventor on US patent (US20190216949) for the AAV6.2FF capsid. D.B.W. has received grant funding, participates in industry collaborations, has received speaking honoraria, and has received fees for consulting, including serving on scientific review committees and board series. Remuneration received by D.B.W. includes direct payments and stock or stock options. D.B.W. also discloses the following paid associations with commercial partners: GeneOne (consultant), Geneos (advisory board), AstraZeneca (advisory board, speaker), Inovio (BOD, SRA, Stock), Sanofi (advisory board) and BBI (advisory board). All other authors declare no completing interests.

## Additional information

[1]The Vaccine and Immunotherapy Center, The Wistar Institute of Anatomy and Biology, Philadelphia, PA 19104, USA. [2]Department of Molecular and Cellular Biochemistry, Indiana University, Bloomington, IN 47405, USA. [3]Inovio Pharmaceuticals, Plymouth Meeting, PA 19462, USA. [4]Public Health Agency of Canada, Winnipeg, MB R3E 3R2, Canada. [5]Vaccines and Immune Therapies, BioPharmaceuticals R&D, AstraZeneca, Gaithersburg, MD 20878, USA. [6]Department of Pathobiology, University of Guelph, Guelph, ON N1G 2W1, Canada. [7]University of Pennsylvania, Philadelphia, PA 19104, USA. [8]Department of Medical Microbiology and Infectious Diseases, University of Manitoba, Winnipeg, MB R3E 0J9, Canada. [9]These authors contributed equally: Jianqiu Du, Ali R. Ali, Katherine Schultheis, Drew Frase, Trevor R. F. Smith. [10]These authors jointly supervised this work: Jean D. Boyer, Mark T. Esser, Jesper Pallesen, Dan W. Kulp, Ami Patel, David B. Weiner. ✉e-mail: dweiner@wistar.org

