## [Peer Review File · Nature Communications]

REVIEWER COMMENTS

Reviewer #1 (Remarks to the Author):

Parzych et al show that an antibody cocktail delivered on plasmids provide protection in rodents against the variants this cocktail is known to be effective against. Overall, the authors show an interesting proof-of-concept for a plasmid-based antibody delivery system against SARS-CoV-2, but they do not realistically state how the emergence of variants impacts the cocktail they chose. Furthermore, the structural analysis does not appear to add much new information, and detracts from the quality of the rest of the manuscript.

MAJOR ISSUES

Main figures & supplement: The raw datapoints for the neutralization assays are not shown in the graphs, so the quality of the fits shown cannot be evaluated. Please show the raw data.

Line 234: Figures 5e and 5f are not discussed or referenced in the text.

Lines 255-264: this analysis is speculative and does not add to the article in a meaningful way, please remove.

Lines 265-308: This structural analysis is reported as if it is the first time these structures have been discovered. These structures have been determined previously elsewhere. Therefore, the authors should discuss differences with these structures, and if these differences could plausibly arise from the different source. Otherwise, this section is highly redundant with the literature and could be removed entirely.

OTHER ISSUES

Line 81: The DMAbs persist for six months, but mAbs alone can last for months in the plasma. It would be useful for readers if the authors would contextualize this difference whenever asserting the superiority of DMAbs. In other words, clarifying the degree to which DNABs are better.

Line 87: Please clarify in the text what – specifically – is meant by ‘first-of-its-class’. What is the ‘class’? What is meant by ‘first’?

Line 108: Intramuscular electroporation sounds painful, do the authors expect this delivery method will become practical for use in a clinic?

Figure 5: The “IC50” value is shown as ng/ml in figure 5e and 5f; should this be ID50? What do the units ng/ml refer to in this context?

Line 231: “This serum also neutralized B.1.1.529/BA.1 pseudotyped virus at ng/ml levels” can the authors explain what “ng/ml levels” mean in the context of serum?

Line 230: Based on what is known in the literature about the 2196/2130 cocktail, would the authors expect “strong” binding to BA.1? Is it possible binding could still occur, but neutralization could not? Please do not conflate binding with neutralization in the text.

Line 280: “rather noisy” – please use specific scientific qualifying remarks.

Line 354: The IgG’s were digested into Fabs before binding to the spike, no? Also, can the authors explain what is meant by “multiple spikes within a timer”? – The spike is the trimer.

Line 334: If the efficacy of the protein or DNA delivery methods are similar, why would anyone opt for the method that entails painful electroporation? This is an opportunity to emphasize what makes the DNA delivery method better.

Line 361: The 2196/2130 cocktail is not effective against BA.1, please change this statement, and adjust the text throughout to account for this. Presumably while the experiments were carried out, variants that emerged could evade the antibody cocktail chosen by the authors. The authors might elect to emphasize how the DNA-delivery method could be modified faster than protein delivery methods, or include many different antibody cocktails.

Reviewer #2 (Remarks to the Author):

Passive antibody administration in the formulation of recombinant proteins has demonstrated protective efficacy in susceptible individuals, but the supply and logistical challenges limit its widespread administration. DNA-delivered neutralizing antibody may significantly simplify the manufacturing process and reduce the medical burden to benefit more underserved populations.

In this work, the authors completed a relatively comprehensive study on DNA-delivered neutralizing antibodies, and found that compared to protein administration, DNA-delivery antibody has equivalent protective efficacy and longer persistent antibody titers in multi murine and hamster models. Moreover, the intermolecular interactions between the cocktail with spike may provide important clues for the development of more promising neutralizing antibodies cocktails.

This work has a great significance for the prevention of COVID-19. There are several points to be addressed before the acceptance of this work.

1. In the last sentence of the introduction, how to understand “expand patient impact” and please replace it with a clearer and unambiguous one.
2. Naked delivery of nucleic acid vaccines or antibodies is notoriously inefficient, the methods to enhance nucleic uptake is important for its application. Please make a brief introduction of “CELLECTRA-EP” in the part of Introduction. In addition, compared with lipid-mediated delivery, what are the advantages and disadvantages of EP?
3. Dramatic antigenic shifts decreased the binding potency of most therapeutic neutralizing antibodies, there`s no need to add a gap on the Y-axis to display a low difference between wide-type and Omicron (Fig 5c and d).
4. The figures of structural details between 2103 and 2196 with RBD needs to be replaced with more refined figures (Fig 7 d-i).
5. Long-term application of single neutralizing antibody is likely of promoting drug-evasion mutants in recipients, and now only one authorized Ly-CoV-1404 (bebtelovimab) could treat both Omicron and its sub-lineages by intravenous injection. From the angle of antibody expression, whether intramuscular injection of the two antibody plasmids at different sites can further improve the antibody titers in vivo, or that is done in this work but not mentioned in the Methods?

Reviewer #3 (Remarks to the Author):

Parzych and colleagues analyzed whether plasmid-encoded antibodies can be employed for SARS-CoV-2 therapy. In brief, they demonstrate that production of antibodies in rodents is efficient and results in potent antiviral activity and show strategies how to improve antibody levels in vivo. Finally, structural insights into how the antibodies inhibit SARS-CoV-2 infection are provided. These results are of interest to the field. Some minor points remain to be addressed:

„and an additional clone, COV2-2381 (2381)22.” It is important to provide a rationale why this antibody was included in the study.

“dual plasmid systems (Supplementary Fig. 1b-c).” Please add half a sentence describing the key features of the dual system.

“These were DNA and RNA optimized to promote in vivo transcript production/processing and inserted,” This reviewer might have overlooked it, but if not already done, these sequences should be made available in the supplement. At least, they should be made available upon request and this should be stated.

“DMAbs 2196(WT) and 2130(WT) were also detected in the bronchoalveolar lavage (BAL)” What about antibody 2381?

On some occasions, the figure legends are incomplete. Please state for each subpanel whether the average of several experiments or a representative experiment is shown. If the former is the case, please state how many experiments were averaged and whether these experiments were carried out with technical replicates, duplicates, triplicates, etc. If the latter is correct, please state the number of technical and biological replicates. Finally, please state whether error bars indicate SD or SEM.

The difference between YTE and control constructs in figure 5A seems minor and any statement regarding differences in PK should be phrased very careful. Further, it should be checked whether differences are statistically significant.

Figures 5e and f should have the same y-axes and should be referenced in the text.

Differences between plasmid-based and protein antibodies are only meaningful if the amounts inoculated into rodents reflect those (on a per kg body weight basis) to be used in humans. Is this the case?

Reviewer Rebuttle: Point-by-point responses to reviewer comments

REVIEWER COMMENTS

Reviewer #1 (Remarks to the Author):

Parzych et al show that an antibody cocktail delivered on plasmids provide protection in rodents against the variants this cocktail is known to be effective against. Overall, the authors show an interesting proof-of-concept for a plasmid-based antibody delivery system against SARS-CoV-2, but they do not realistically state how the emergence of variants impacts the cocktail they chose. Furthermore, the structural analysis does not appear to add much new information, and detracts from the quality of the rest of the manuscript.

- 1. *Main figures & supplement: The raw datapoints for the neutralization assays are not shown in the graphs, so the quality of the fits shown cannot be evaluated. Please show the raw data.***

Neutralization curves were represented as best-fit lines to visually simplify the data. However, we are happy to add the individual data points for each curve at the reviewer's request.

All neutralization curves have been updated to include individual datapoints used to define the best-fit lines (Fig 1c; Fig 2a; Fig 3q; Fig 5b,e-f; Fig S2b; Fig S3c, Fig S4c; FigS5)

- 2. *Line 234: Figures 5e and 5f are not discussed or referenced in the text.***

References to Figure 5e (pg 11; line 239) and Figure 5f (pg 11; line 242) have been added and are described in the text.

- 3. *Lines 265-308: This structural analysis is reported as if it is the first time these structures have been discovered. These structures have been determined previously elsewhere. Therefore, the authors should discuss differences with these structures, and if these differences could plausibly arise from the different source. Otherwise, this section is highly redundant with the literature and could be removed entirely.***

The reviewer is correct; structural information for recombinant mAbs COV2-2196 and COV2-2130 has been published (Zost et al. Nature. 2020; Dong et al. Nature Microbiology. 2021). These studies were acknowledged and cited in the discussion section (pgs 15-16; lines 348-357).

While ours are not the first structures of these clones, there are several aspects that make our structural data/ interpretation unique:

1. *In vivo*-launched Fabs:

- This is the first structure of nucleic acid-encoded dFabs produced *in vivo*; structural comparisons between bioprocessed and *in vivo*-expressed Fabs have not been conducted.
- Our data demonstrate that plasmid-derived Fabs produced within the host are structurally, physically and functionally comparable to standard protein-based mAb drug products.
- These preclinical studies comparing DMAbs to the current clinical standard of care (protein IgG) further validates our approach.

2. Use of native-like SARS-CoV-2 Spike trimer (antigen)

- In contrast to the previously solved structures in which mAbs COV2-2196 and COV2-2130 were complexed with SARS-CoV-2 Spike-RBD (Dong et al. Nature Microbiology. 2021), the antigen developed for our cryo-EM studies was a stabilized SARS-CoV-2 trimer with high integrity and native-like confirmation
- To our knowledge, this is the first high-resolution structural determination of these clones in complex with trimeric spike, allowing us to observe the binding of multiple copies of each dFab as is likely to occur during live infection
- This offers a better understanding of the cooperative binding effects that take place at high Fab occupancy

4. *Lines 255-264: this analysis is speculative and does not add to the article in a meaningful way, please remove.*

The section (pg 12/ lines 263-272) in question describes the physical distances measured between bound dFabs complexed with SARS-CoV-2 spike trimer.

As this is the first study to describe clones COV2-2196 and COV2-2130 complexed with trimeric spike with high-resolution, it is the first opportunity to observe and measure the spatial arrangement of multiple dFab copies bound to a single trimer. Proximity calculations support the occurrence of dFab-to-dFab as well as the potential for Fc-to-Fc interactions, which is in line with the agglutination that is known to occur upon IgG complexation.

5. *Line 81: The DMAbs persist for six months, but mAbs alone can last for months in the plasma. It would be useful for readers if the authors would contextualize this difference whenever asserting the superiority of DMAbs. In other words, clarifying the degree to which DNABs are better.*

The statement in question (pg 4; line 85) states that we observed “*the persistence of 2130 and 2196 DMAbs for over six months*” following plasmid delivery. This is a description of DMAb expression and does not include any comparisons.

6. *Line 87: Please clarify in the text what – specifically – is meant by ‘first-of-its-class’. What is the ‘class’? What is meant by ‘first’?*

The indicated statement (pg 4; line 91) reads: “Furthermore, first-of-its-class structural assessment of *in vivo*-launched 2196 and 2130 DMAbs was performed using cryo-EM.”

We have removed this phrase from the text for clarity (pg 4; line 91).

7. Line 108: Intramuscular electroporation sounds painful, do the authors expect this delivery method will become practical for use in a clinic?

We updated the introduction to include additional information about electroporation as a delivery strategy, ensuring readers that this has been validated in patients (pgs 3-4; lines 72-77).

8. Figure 5: The “IC50” value is shown as ng/ml in figure 5e and 5f; should this be ID50? What do the units ng/ml refer to in this context?

We are happy to clarify this distinction.

For all neutralization data, the line of best fit defines the inhibitory dilution that yields 50% viral neutralization compared to control wells (ID50). However, these values are strongly affected by circulating antibody concentrations. To control for differences in *in vivo* DMAb expression levels, we quantified the concentration of human IgG (DMAb) in each sera sample (ng/mL) then used the ID50 to calculate the inhibitory concentration that yields 50% viral neutralization (IC50) according to the following equation: $IC50 = \text{DMAb titer} / ID50$. This allows us to compare the relative molecular potency of these antibodies independent of expression levels.

These values/calculations are described in Methods under *Neutralization Assay: Pseudotyped virus* section (pg. 24; lines 542-554).

9. Line 231: “This serum also neutralized B.1.1.529/BA.1 pseudotyped virus at ng/ml levels” can the authors explain what “ng/ml levels” mean in the context of serum?

Antiviral activity of these DMAbs against the BA.1 sublineage was performed using a pseudoviral neutralization assay. The ID50 value generated for each serum sample was used in conjunction with the DMAb serum titer to calculate the concentration of antibody that corresponds to this 50% neutralization (see comment #8 above).

In this case, we observed that the DMAb cocktail mediated BA.1 neutralization at serum DMAB concentrations of ~ 200 ng/mL (Fig. 5e; pg11; lines 239-241).

10. Line 230: Based on what is known in the literature about the 2196/2130 cocktail, would the authors expect “strong” binding to BA.1? Is it possible binding could still occur, but neutralization could not? Please do not conflate binding with neutralization in the text.

Several publications have reported significant, though somewhat reduced, activity of the 2130/2196 mAb cocktail against the Omicron lineages, including BA.1 (see #14 below for citations). Based on these reports, we did expect to observed reactivity with the Omicron trimer in our assays.

We agree with the reviewer that binding is not necessarily indicative of neutralization. We therefore evaluated both binding to (via ELISA) and neutralization (via pseudovirus assay) of BA.1 by the *in vivo*-expressed DMAB cocktail. The DMAb cocktail recognized recombinant BA.1 trimer (Figure 5c-d) and exhibited neutralizing activity against BA.1 pseudovirus (Figure 5e).

We added the comment “*albeit at reduced potency compared to the USA-WA1/2020 strain*” (pg 11; lines 240) to acknowledge the somewhat reduced activity against some Omicron lineages. To further support these findings, we included recent relevant citations (pg 11; lines 243) and a reference to the updated FDA guidelines on Evusheld administration (pg 11; lines 244).

11. Line 280: “rather noisy” – please use specific scientific qualifying remarks.

As requested, we removed this phrase and replaced it with the following statement for clarity (pg 13; lines 287-291):

“Here we observed inherent flexibility (higher than average B factor) of complexes containing the 2196 Fab alone, both in regard to movement of the RBD itself and its interaction with the 2196 Fab (Figure 6a-c). In contrast, simultaneous binding of both 2196/2130 Fabs results in an ordered and well-defined complex with reduced motion/ flexibility, signifying stabilizing interactions between bound Fabs (Figure 6d-f).”

12. Line 354: The IgG’s were digested into Fabs before binding to the spike, no? Also, can the authors explain what is meant by “multiple spikes within a timer”? – The spike is the trimer.

Yes, the reviewer is correct. As described in the results (pg 11; lines 249-252) and methods (pg 27; lines 615-619) sections, serum-derived DMAbs were digested into Fabs prior to complex formation. To clarify this, we employed the more accurate term of *DNA-encoded Fabs* (dFabs) throughout the manuscript when referring to the structural complexes.

Regarding the trimer description, adjusted the sentence to read:

“Our high-resolution cryo-EM of the full trimeric 2130/2196/S complex revealed the concurrent binding of multiple copies of each dFab, allowing us to visualize and measure the proximity of bound dFabs at nearly full trimer occupancy (5/6 binding sites). Measurements of physical distances support a basis for multiple IgG-to-IgG interactions within the trimer” (pg 16; lines 359-363):

13. Line 334: If the efficacy of the protein or DNA delivery methods are similar, why would anyone opt for the method that entails painful electroporation? This is an opportunity to emphasize what makes the DNA delivery method better.

As suggested, we added the following statement in the discussion that emphasizes the value of DNA-delivery (pg 16; lines 371-377):

“As a supplement to traditional protein IgG, a DNA-delivery could improve the availability of such mAb products by addressing challenges typically associated with the large-scale production, distribution and cold-chain storage of bioprocessed biologics. This could help extended access to underserved populations that may be otherwise restricted due to logistical and/or financial restraints as well as potentially avoid supply chain limitations in the context of future pandemics. Further development and optimization of this technology is likely important.”

14. Line 361: The 2196/2130 cocktail is not effective against BA.1, please change this statement, and adjust the text throughout to account for this. Presumably while the experiments were carried out, variants that emerged could evade the antibody cocktail chosen by the authors. The authors might elect to emphasize how the DNA-delivery method could be modified faster than protein delivery methods, or include many different antibody cocktails.

Numerous studies have demonstrated the continued activity of this cocktail against BA.1, albeit with reduced potency compared to earlier variants (*VanBlargan et al. Nat Med. 2022; Dejnirattisai et al. BioRxiv. 2021.; Tada et al. eBioMedicine. 2022.*).

Furthermore, activity is largely regained against BA.2 (*Takashita et al. NEJM. 2022.; Case et al. Nat Comm. 2022.*) and subsequent sublineages BA.4/ BA.5 (*Yamasoba et al. Lancet Infect. Dis. 2022.; Tuekprakhon et al. Cell. 2022.*)

Consistent with these observations, AZD7442 (tixagevimab/cilgavimab cocktail) has retained authorization for clinical use against all current variants, though at an increased dosage following the FDA’s revised guidelines (*“FDA authorizes revisions to Evusheld dosing (Update 2/24/2022)”*); [https://www.fda.gov/drugs/drug-safety-and-availability/fda-authorizes-revisions-evusheld-dosing.](https://www.fda.gov/drugs/drug-safety-and-availability/fda-authorizes-revisions-evusheld-dosing)

This is in sharp contrast to several other mAb products for which EUA was revoke due to inactivity against the Omicron lineages, including REGEN-COV (casirivimab/imdeviab), the bamlanivimab/ etesevimab cocktail and monotherapy sotrovimab. AZD7442 and bebtelovimab remain the only mAb products approved for unrestricted clinical use.

We included additional citations/ references throughout the text describing the activity of 2196/2130 mAbs against Omicron variants.

Reviewer #2 (Remarks to the Author):

Passive antibody administration in the formulation of recombinant proteins has demonstrated protective efficacy in susceptible individuals, but the supply and logistical challenges limit its widespread administration. DNA-delivered neutralizing antibody may significantly simplify the manufacturing process and reduce the medical burden to benefit more underserved populations.

In this work, the authors completed a relatively comprehensive study on DNA-delivered neutralizing antibodies, and found that compared to protein administration, DNA-delivery antibody has equivalent protective efficacy and longer persistent antibody titers in multi murine and hamster models. Moreover, the intermolecular interactions between the cocktail with spike may provide important clues for the development of more promising neutralizing antibodies cocktails.

This work has a great significance for the prevention of COVID-19. There are several points to be addressed before the acceptance of this work.

15. In the last sentence of the introduction, how to understand “expand patient impact” and please replace it with a clearer and unambiguous one.

This line in the abstract (pg 2; line 44-45) was replaced with the following:

“These data support the further study of DMAb technology in the development and delivery of valuable biologics.”

16. Naked delivery of nucleic acid vaccines or antibodies is notoriously inefficient, the methods to enhance nucleic uptake is important for its application. Please make a brief introduction of “CELLECTRA-EP” in the part of Introduction. In addition, compared with lipid-mediated delivery, what are the advantages and disadvantages of EP?

As requested, we have incorporated the following statement in the introduction (pg 3; lines 72-76):

“Avoiding the complex production and limited stability typically associated with protein or lipid-based formulations, this platform allows the efficient delivery of temperature-stable, purified DNA using clinically-validated electroporation technology (CELLECTRA-EP; Inovio Pharmaceuticals) to facilitate uptake and expression”

17. Dramatic antigenic shifts decreased the binding potency of most therapeutic neutralizing antibodies, there`s no need to add a gap on the Y-axis to display a low difference between wide-type and Omicron (Fig 5c and d).

The Y-axes have been adjusted as requested for Figures 5c and 5d.

18. The figures of structural details between 2103 and 2196 with RBD needs to be replaced with more refined figures (Fig 7 d-i).

All panels in Figure 7 were checked for accuracy, ensuring that the interactions of interest are clearly visible to the readers.

19. Long-term application of single neutralizing antibody is likely of promoting drug-evasion mutants in recipients, and now only one authorized Ly-CoV-1404 (bebtelovimab) could treat both Omicron and its sub-lineages by intravenous injection. From the angle of antibody expression, whether intramuscular injection of the two antibody plasmids at different sites can further improve the antibody titers in vivo, or that is done in this work but not mentioned in the Methods?

The methods section (*Animals, in vivo DMAb delivery and sample collection*) was updated to include the following statement (pg 19; Line 421):

“In animals receiving both DMAbs, plasmids for each clone were injected at separate sites.”

Reviewer #3 (Remarks to the Author):

Parzych and colleagues analyzed whether plasmid-encoded antibodies can be employed for SARS-CoV-2 therapy. In brief, they demonstrate that production of antibodies in rodents is efficient and results in potent antiviral activity and show strategies how to improve antibody levels in vivo. Finally, structural insights into how the antibodies inhibit SARS-CoV-2 infection are provided. These results are of interest to the field. Some minor points remain to be addressed:

20. „and an additional clone, COV2-2381 (2381)22.” It is important to provide a rational why this antibody was included in the study.

We included the following statement to contextualize the inclusion of DMAb 2381 in the early studies (pg 5; lines 106-108):

“With properties similar to COV2-2196, mAb COV2-2381 also mediated protection against SARS-CoV-2 in large animal models and was therefore of interest.”

21. “dual plasmid systems (Supplementary Fig. 1b-c).” Please add half a sentence describing the key features of the dual system.

We have added the following sentence to the results section as requested (pg 5; lines 110-112):

“These were generated as single or dual plasmid designs in which the heavy chains and light chains for each clone were encoded on the same (single) or separate (dual) plasmids (Supplementary Fig. 1b-c).”

- 22. “These were DNA and RNA optimized to promote in vivo transcript production/processing and inserted,” This reviewer might have overlooked it, but if not already done, these sequences should be made available in the supplement. At least, they should be made available upon request and this should be stated.**

The structures for S/2196 and S/2196/2130 complexes have been deposited in PDB (codes 8D8R and 8D8Q, respectively) and EMDB (codes EMD-27255 and EMD-27254, respectively), thereby making the sequences of these constructs publicly accessible.

Accession codes were added to Supplementary Table 1 and the Data Availability section (pg 29; lines 660-664) of the manuscript.

- 23. “DMAbs 2196(WT) and 2130(WT) were also detected in the bronchoalveolar lavage (BAL)” What about antibody 2381?**

We did not investigate the levels of DMAb 2381 in the BAL following plasmid delivery, as we had begun to focus on the 2130/2196 pair for further development.

- 24. On some occasions, the figure legends are incomplete. Please state for each subpanel whether the average of several experiments or a representative experiment is shown. If the former is the case, please state how many experiments were averaged and whether these experiments were carried out with technical unicates, duplicates, triplicates, etc. If the latter is correct, please state the number of technical and biological replicates. Finally, please state whether error bars indicate SD or SEM.**

All figure legends have been updated to include this information, as applicable.

- 25. The difference between YTE and control constructs in figure 5A seems minor and any statement regarding differences in PK should be phrased very careful. Further, it should be checked whether differences are statistically significant.**

We adjusted the results section to clarify these findings, as follows:

“YTE-containing constructs display a modestly improvement in titers compared to their non-YTE counterparts, though this does not reach statistical significance. This is the first study to combine in vivo production and YTE function, both of which appear to contribute to sustained DMAb expression to different degrees.” (pg 10; lines 226-228)

26. Figures 5e and f should have the same y-axes and should be referenced in the text.

The axes for Figs. 5e-f have been adjusted as requested. References to these figure panels have been added to the results section (pg 11; lines 241-242).

27. Differences between plasmid-based and protein antibodies are only meaningful if the amounts inoculated into rodents reflect those (on a per kg body weight basis) to be used in humans. Is this the case?

New gene-based approaches traditionally are studied in animal models prior to potential advancement into the clinic, where such dose relationships can be determined. The field looks forward to being able to address this question. As such, this study could be considered a guide-post.